



# Stratospheric Ozone Response to Sulfate Aerosol and Solar Dimming Climate Interventions based on the G6 Geoengineering Model Intercomparison Project (GeoMIP) Simulations

Simone Tilmes[1], Daniele Visioni[2], Andy Jones[3], James Haywood[3], Roland Séférian[4], Pierre Nabat[4], Olivier Boucher[5], Ewa Monica Bednarz[2], and Ulrike Niemeier[6]

[1]Atmospheric Chemistry, Observations, and Modeling Laboratory, National Center for Atmospheric Research, Boulder, CO, USA
[2]Sibley School for Mechanical and Aerospace Engineering, Cornell University, Ithaca, NY, USA
[3]Met Office Hadley Centre, Exeter, EX1 3PB, UK
[4]CNRM, Université de Toulouse, Meteo-France, CNRS, Toulouse, France
[5]Institut Pierre-Simon Laplace, Sorbonne Université/CNRS, Paris, France
[6]Max Planck Institute for Meteorology, Hamburg, Germany

**Correspondence:** Simone Tilmes (tilmes@ucar.edu)

**Abstract.** This study assesses the impacts of sulfate aerosol intervention (SAI) and solar dimming on stratospheric ozone based on the G6 Geoengineering Model Intercomparison Project (GeoMIP) experiments, called G6sulfur and G6solar. For G6sulfur the stratospheric sulfate aerosol burden is increased to reflect some of the incoming solar radiation back into space in order to cool the surface climate, while for G6solar the global solar constant is reduced to achieve the same goal. The high

emissions scenario SSP5-8.5 is used as the baseline experiment and surface temperature from the medium emission scenario SSP2-4.5 is the target. Based on three out of six Earth System Models (ESMs) that include interactive stratospheric chemistry, we find significant differences in the ozone distribution between G6solar and G6sulfur experiments compared to SSP5-8.5 and SSP2-4.5, which differ by both region and season. Both SAI and solar dimming methods reduce incoming solar insolation and result in tropospheric temperatures comparable to SSP2-4.5 conditions. G6sulfur increases the concentration of absorbing

sulfate aerosols in the stratosphere, which increases lower tropical stratospheric temperatures by between 5 to 13 K for six different ESMs, leading to changes in stratospheric transport. The increase of the aerosol burden also increases aerosol surface area density, which is important for heterogeneous chemical reactions. The resulting changes in ozone include a significant reduction of total column ozone (TCO) in the Southern Hemisphere polar region in October of 10 DU at the onset and up to 20 DU by the end of the century. The relatively small reduction in TCO for the multi-model mean in the first two decades results

from variations in the required sulfur injections in the models and differences in the complexity of the chemistry schemes, with no significant ozone loss for 2 out of 3 models. The decrease in the second half of the $21^{st}$ century counters increasing TCO between SSP2-4.5 and SSP5-8.5 due to the super-recovery resulting from increasing greenhouse gases. In contrast, in the Northern Hemisphere (NH) high latitudes, only a small initial decline in TCO is simulated, with little change in TCO by the end of the century compared to SSP5-8.5. All models consistently simulate an increase in TCO in the NH mid-latitudes

up to 20 DU compared to SSP5-8.5, in addition to 20 DU increase resulting from increasing greenhouse gases between SSP2-





4.5 and SSP5-8.5. G6solar counters zonal wind and tropical upwelling changes between SSP2-4.5 and SSP5-8.5 but does not change stratospheric temperatures. Solar dimming results in little change in TCO compared to SSP5-8.5 and does not counter the effects of the ozone super-recovery. Only in the tropics, G6solar results in an increase of TCO of up to 8 DU compared to SSP2-4.5, which may counter the projected reduction due to climate change in the high forcing future scenario. This work

identifies differences in the response of SAI and solar dimming on ozone, which are at least partly due to differences and shortcomings in the complexity of aerosol microphysics, chemistry, and the description of ozone photolysis in the models. It also identifies that solar dimming, if viewed as an analog to SAI using a predominantly scattering aerosol, would, for the most part, not counter the potential harmful increase in TCO beyond historical values induced by increasing greenhouse gases.

## 1   Introduction

There has been an increasing interest in researching Climate Intervention (CI) strategies because even ambitious mitigation efforts may not be sufficient to keep global mean temperature targets below $1.5°C$ above pre-industrial, as needed to prevent more serious climate impacts (Masson-Delmotte et al., 2021). Furthermore, current commitments to reduce greenhouse gases emissions are falling way short of reaching the required temperature targets. Therefore, CI strategies beyond mitigation and adaptation may be the only way to prevent severe impacts on society and ecosystems. While research has been increasing in

this direction, there is still considerable uncertainty on the effects of different proposed CI proposals on the Climate System. One of the most studied Solar Radiation Management (SRM) approaches is the continuous injection of sulfur ($SO_2$) into the stratosphere, which results in an enhanced stratospheric aerosol burden that reflects some of the incoming sunlight to space and therefore cools the Earth's surface (NAS, 2021). This approach is called Stratospheric Aerosol Intervention (SAI) in the following. An often-mentioned concern of SAI is the impact on stratospheric ozone, particularly the delay of the Antarctic

ozone recovery (e.g., Tilmes et al., 2008; Heckendorn et al., 2009). The increase of stratospheric surface area density (SAD) from SAI is expected to impact heterogeneous chemical reactions similar to the observed impacts after large volcanic eruptions (e.g., Solomon, 1999). In addition, the heating of the lower tropical stratosphere from sulfate aerosols causes changes in stratospheric transport and circulation (e.g., Niemeier and Schmidt, 2017; Richter et al., 2017). Both these changes impact stratospheric ozone (e.g., Tilmes et al., 2017).

Recent studies investigated the impacts of SAI on stratospheric ozone with simulations using the Community Earth System Model (CESM) with the Whole Atmosphere Community Climate Model (WACCM) as the atmospheric component, a configuration dented by CESM(WACCM). The experiments employed a high climate forcing future scenario (using Representative Concentration Pathway, RCP8.5 emissions) and required continuously increasing sulfur injections to keep surface temperatures at 2020 conditions (Tilmes et al., 2021). This study found that even a transient phase-in of sulfur injections can significantly

deepen the ozone hole over Antarctica in October within the first ten years of the application. Furthermore, the heating of the tropical lower stratosphere results in an increase in total column ozone in mid-to-high latitudes in the Northern Hemisphere winter. Another study analyzed the effects of SAI on ozone using two different baseline scenarios, the Shared-Socioeconomic Pathway (SSP) high forcing scenario SSP5-8.5 and the SSP5-3.4-OS scenario (Tilmes et al., 2020). SSP5-3.4-OS follows





SSP5-8.5 until 2040 and afterwards assumes substantial decarbonization and active removal of $CO_2$ from the atmosphere,
resulting in an overshoot (OS) of threshold temperatures defined by the Paris Agreement. SAI was applied in both scenarios
to maintain temperatures below 1.5°C or 2°C, with the latter allowing a phase-out of sulfur injection once greenhouse gases
concentrations in the atmosphere have started to decline in a so-called "peak-shaving" scenario (Wigley, 2006; Tilmes et al.,
2016; MacMartin and Kravitz, 2016; Masson-Delmotte et al., 2018)

The impact of SAI on ozone depends on the increase in SAD and aerosol mass that increases with increasing $SO_2$ injection
amount (Tilmes et al., 2020). A baseline scenario with higher climate forcings that requires much larger sulfur injections to
reach target surface temperatures by the end of the century, resulted in a much stronger impact on ozone (both increase and
decrease depending on the region and season) than a scenario that would phase out injections towards the end of the $21^{st}$
century. However, it is unclear how representative these recent studies are since they only used one modeling framework,
CESM(WACCM).

The Geoengineering Model Intercomparison Project (GeoMIP) has defined a standardized set of model experiments to assess
the effects of SRM methods, including SAI (Kravitz et al., 2011, 2015). Some of the earlier experiments include the injections
of sulfur into the stratosphere, such as the G3 and G4 experiments (Pitari et al., 2014; Xia et al., 2017). These earlier modeling
experiments used the Climate Model Intercomparison Project 5 (CMIP5) scenario RCP4.5. They considered either a constant
equatorial injection of 8 Tg $SO_2$/yr between 2020 and 2070 (G4) or a progressively increasing injection of $SO_2$ to maintain
temperatures at 2020 levels (G3). The injection altitude was different among models. Only four models included the required
processes to simulate the impacts of SAI on ozone for the G4 experiment and two for the G3 experiment. Pitari et al. (2014)
found a decline of stratospheric ozone in the polar regions with sulfur injections for all the participating models. However, the
stratospheric aerosol distributions in those models presented considerable differences, making conclusions about the overall
impacts of SAI on ozone hard to determine.

More recent GeoMIP experiments, G6sulfur and G6solar, defined for CMIP6 future emission pathways (SSPs), were de-
signed to explore the effects of SAI and solar dimming in a more policy-relevant setting (Kravitz et al., 2015). Both experiments
employ SSP5-8.5 as the baseline scenario. G6sulfur requires the application of sulfur injections between 10S° – 10°N in lati-
tude and around 18-20 km altitude to keep surface temperatures at the same values as those simulated in the SSP2-4.5 scenario
for the 2020-2100 period (the target scenario). G6solar requires reducing the global solar constant to offset the same temper-
ature difference to reach SSP2-4.5 values. The purpose of comparing both G6sulfur and G6solar is to identify differences in
those approaches, as past analyses have often described the reduction in the solar constant as a proxy for SAI (Ban-Weiss and
Caldeira, 2010; Irvine et al., 2019). However, Niemeier et al. (2013) have shown that climate impacts, especially precipitation,
differ between SAI and solar dimming. Visioni et al. (2021a) have shown that differences between these applications were iden-
tified regarding surface climate impacts and their effect on ozone, even if the solar dimming was applied to achieve the same
global mean, inter-hemispheric and pole-to-Equator surface temperature targets as SAI. Similarly, Xia et al. (2017) outlined
differences in the effects of solar dimming and SAI on stratospheric and tropospheric ozone. Both these earlier studies used
CESM(WACCM), while G6sulfur and G6solar experiments have been performed by six ESMs. Another proposal suggests us-
ing aerosols for SAI that absorb less solar radiation when integrated across the solar spectrum (Keith and Irvine, 2016; Dykema





et al., 2016), which may reduce some of the climate impacts, including the precipitation reduction over southern Europe in
winter (Jones et al., 2021) and weakening of the monsoonal precipitation over India (Simpson et al., 2019). Therefore, solar
dimming may be a closer analog to SAI approaches using less absorbing aerosols than sulfates.

This study explores the impacts of SAI and global solar dimming on stratospheric ozone based on the G6sulfur and G6solar
GeoMIP experiments. In total, the results of six Earth System Models (ESMs) that performed these GeoMIP experiments
are available. However, only three different ESMs include comprehensive interactions between chemistry and aerosols in the
stratosphere. Section 2 describes details of the experimental design and models participating in this study. Results are described
in Section 3 and include changes in stratospheric temperatures and transport and surface area distribution for both G6solar and
G6sulfur, and the effects on ozone concentration total column ozone (TCO) for selected regions and seasons. A summary is
given in Section 4, and discussion and conclusions are presented in Section 5.

## 2   Experimental Design and Model Description

The GeoMIP G6solar and G6sulfur experiments use the SSP5-8.5 high greenhouse gas forcing scenario as their baseline
scenario. SAI or global solar dimming are applied to reduce global surface temperatures to the levels derived for the SSP2-4.5
for each model. The experiment does therefore not aim towards reaching surface temperature targets of $1.5°C$. The annual
forcing required to achieve this goal in these experiments depends on the surface air temperature difference between SSP5-8.5
and SSP2-4.5. This difference strongly increases in the second half of the $21^{st}$ century, with $0.19 \pm 0.04$ K global-mean surface
temperature differences in 2040 and $0.62 \pm 0.05$ K in 2060, $1.46 \pm 0.14$ K in 2080 and $2.42 \pm 0.22$ K in 2100 based on the
GeoMIP multi-model mean in the participating models considering a 10-year running mean (Visioni et al., 2021b). According
to these differences, the models required much less solar dimming or aerosol increase in the first half of the century than the
second half to reach the surface temperature of the target (SSP2-4.5) experiment.

Six models participated in these experiments, as listed in Table 1 (adapted from Visioni et al. (2021b)). The G6sulfur
experiment required sulfur injections directly into the stratosphere. Only three models that performed G6 experiments include
an interactive aerosol microphysical model in the stratosphere. Two of these three models (IPSL-CM6A-LR and UKESM1-0-
LL) injected $SO_2$ uniformly between $10°N$ and $10°S$ and 18 and 20 km of altitude at a single longitude ($0°$). These models
used a "distinct stepping" of injections every ten years. CESM2-WACCM6 injected $SO_2$ at the Equator at 25 km altitude.
CESM2(WACCM) used a feedback-control algorithm (MacMartin et al., 2017) to identify the injection amount required every
year to reach the target surface temperatures. The other models used precalculated aerosol distributions to prescribe aerosol and
optical properties, where a prescribed aerosol distribution was scaled to reach the required target temperature. CNRM-ESM2-1
used an input dataset provided by GeoMIP (from the G4SSA experiment (Tilmes et al., 2015)), while MPI-ESM prescribed
their aerosol distribution derived from the aerosol microphysical simulations described in Niemeier and Schmidt (2017), and
Niemeier et al. (2020).

Only three out of the six coupled Earth System models, UKESM1-0-LL, CESM2(WACCM), and CNRM-ESM2-1 included
interactive stratospheric chemistry, including ozone and water vapor coupled to the radiation scheme, which is required to





**Table 1.** Summary of model simulations used in this work. Adapted from Visioni et al. (2021b).

| Model name | SSP2-4.5 | SSP5-8.5 | G6solar | G6sulfur | Stratospheric aerosols in G6sulfur | Interactive stratospheric ozone |
|---|---|---|---|---|---|---|
| | (number of ensemble members) | | | | | |
| **CESM2(WACCM)** | 2 | 2 | 2 | 2 | $SO_2$ injection | Yes |
| **CNRM-ESM2-1** | 3 | 3 | 1 | 3 | AOD scaled | Yes |
| **IPSL-CM6A-LR** | 1 | 1 | 1 | 1 | $SO_2$ injection | No |
| **MPI-ESM1.2-LR** | 3 | 3 | 3 | 3 | AOD scaled | No |
| **MPI-ESM1.2-HR** | 3 | 3 | 3 | 3 | AOD scaled | No |
| **UKESM1-0-LL** | 3 | 3 | 3 | 3 | $SO_2$ injection | Yes |

determine the impacts of SAI on ozone. Only two out of the three models, UKESM1-0-LL and CESM2(WACCM), include interactive aerosol microphysical schemes. The other three models used prescribed ozone fields, which differed only between SSP5-8.5 and SSP2-4.5 (Keeble et al., 2021). The CNRM-ESM2-1 chemistry scheme considers 168 chemical reactions, among
which 39 are photolysis reactions, and 9 reactions that represent heterogeneous chemistry. This scheme is applied above 560 hPa but does not include non-methane hydrocarbon chemistry in the calculation of tropospheric ozone. The model does not include an interactive aerosol microphysical model in the stratosphere and uses a prescribed stratospheric aerosol distribution. The photolytic calculation consideres changes in the chemical composition, but does not consider changes in aerosols. A full description and evaluation of CNRM-ESM2-1 can be found in Séférian et al. (2019). The evaluation of the ozone radiative
forcing is described in Michou et al. (2020).

The UKESM1-0-LL model uses a combined stratospheric-tropospheric chemistry scheme (Archibald et al., 2020), including 84 tracers, 199 bimolecular reactions, 25 unimolecular and termolecular reactions, 59 photolytic reactions, five heterogeneous reactions, and three aqueous-phase reactions for the sulfur cycle from the United Kingdom Chemistry and Aerosol (UKCA) model. Although an extended stratospheric chemistry scheme is available that includes explicit treatment of most of the long-
lived ODS of importance for the recovery of stratospheric ozone and participated in CCMI (e.g., Dhomse et al., 2018), this scheme was not used in UKESM1-0-LL. Instead, the lower boundary conditions of halogenated ODSs are "lumped" into three main halogenated source gases (CFC11, CFC12 and CH3Br). UKESM1-0-LL uses the UKCA-GLOMAP modal aerosol scheme (Mann et al., 2010) and interactive Fast-JX photolysis scheme, which is applied to derive photolysis rates between 177 and 850 nm, as described in Telford et al. (2013). In the lower mesosphere, photolysis rates are calculated using look-up tables
(Lary and Pyle, 1991). The performance of UKESM1-0-LL is described in detail in Sellar et al. (2019).

CESM2-WACCM6 uses the Whole Atmosphere Community Climate Model version 6 (WACCM6) as its atmosphere component. The model includes comprehensive chemistry in the troposphere, stratosphere, mesosphere, and lower thermosphere (TSMLT), including 231 species, 150 photolysis reactions, 403 gas-phase reactions, 13 tropospheric heterogeneous reactions, and 17 stratospheric heterogeneous reactions (Emmons et al., 2020). The photolytic calculations use both inline chemical
modules and a lookup table approach, which does not consider changes in aerosols. CESM2-WACCM6 includes prognostic





representation of stratospheric aerosols based on sulfur emissions from volcanoes and other sources (Mills et al., 2017). The performance of CESM2-WACCM6 is described in detail in Gettelman et al. (2019).

The three models described above all participated in the CMIP6 and evaluations of stratospheric ozone and water vapor showed generally good agreement with observations, but a few notable differences (Keeble et al., 2021). In comparison to observations and the multi-model mean, UKESM1-0-LL significantly overestimated total column ozone globally. This behavior was partially related to the limited treatment of heterogeneous chlorine and bromine chemistry. The model produces a more negative trend in high latitudes than observed between 1960 and 2014. CNRM-ESM2-1 underestimates TCO in the polar regions while overestimating TCO in the tropics but shows a reasonable decline in ozone between 1960 and 2014 (Séférian et al., 2019). CESM2-WACCM6 TCO is in good agreement with observations but underestimates the negative trend between 1960 and 2014 in the Northern Hemisphere high latitudes. In the following analyses, we show TCO model results relative to 2020 conditions to remove model biases in TCO.

## 3 Results

Overall changes in stratospheric ozone concentration are due to a combination of different factors: dynamical changes resulting in transport differences, changes in heterogeneous chemistry induced by larger SAD values, temperature- and photolysis-driving differences in reaction rates (Pitari et al., 2014; Tilmes et al., 2018; Visioni et al., 2021a), as outlined in this section.

### 3.1 Effects of SAI and solar dimming on atmospheric temperature and winds

In G6sulfur and G6solar, SAI and solar dimming have been applied to the SSP5-8.5 baseline scenario to reach global-mean surface temperatures simulated for the target scenario SSP2-4.5. To understand the stratospheric changes from the G6 application, we first discuss changes between the baseline and the target scenario. The increase in greenhouse gases results in an increase in tropospheric temperatures and cooling of the stratosphere (Fels et al., 1980; Pisoft et al., 2021). This will introduce changes in the meridional gradient of zonal-mean temperature, which by geostrophic balance imply changes in zonal-mean wind. Changes in the zonal wind, especially near the top and Equatorward flanks of the tropospheric jets, then affect wave forcing and with that the shallow-branch of the Brewer-Dobson Circulation (BDC). In addition, changes in the tropical upwelling and a strengthening of the polar vortex can be inferred for the three models between SSP2-4.5 and SSP5-8.5 which are however not significant (Figure A1, middel panels).

G6sulfur and G6solar act in addition to the changes caused by greenhouse gases. The increase Aerosol Optical Depth (AOD) and solar dimming successfully counters the temperature increase in the troposphere and at the global surface between SSP2-4.5 and SSP5-8.5 for all the models (Visioni et al., 2021b). However, stratospheric temperature changes from high greenhouse gas concentrations in the baseline scenario SSP5-8.5 are not reversed. For G6sulfur, the increased stratospheric sulfate burden with increasing sulfur injections causes warming of the lower tropical stratosphere compared to the baseline experiment SSP5-8.5 (Figure 1) and compared to SSP2-4.5 (Figure A2).



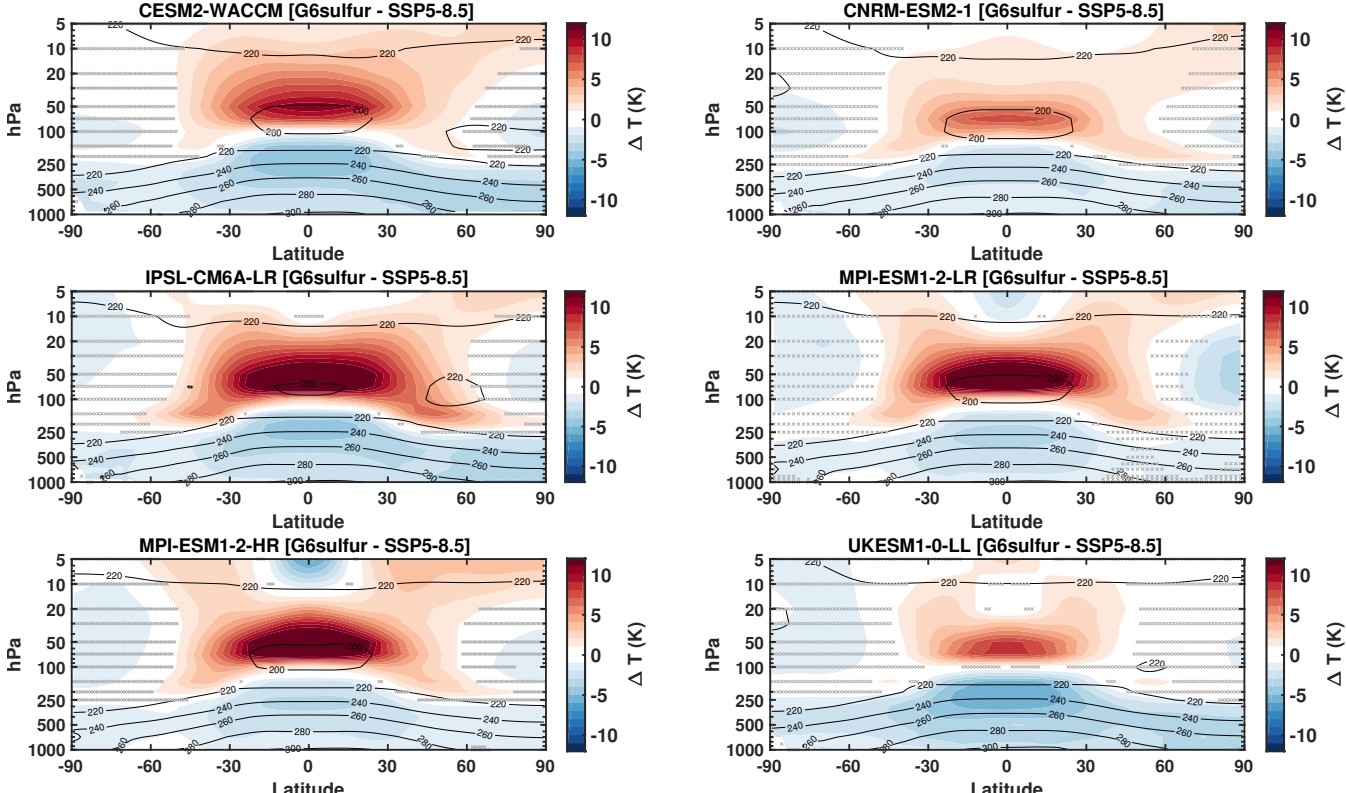

**Figure 1.** Zonal mean temperature changes (2080-99) between G6sulfur and SSP5-8.5 for all GeoMIP models participating in the G6 experiment. Contour lines show the baseline SSP5-8.5 temperature. Grey horizontal lines indicate changes that are not statistically significant over the time period considered using a double-sided t-test at 95% confidence levels.

Different models substantially differ in the amount of stratospheric heating in G6sulfur by the end of the experiment (2080-2099). This difference results from variations in the amount of aerosol mass and size, aerosol distribution, and the specifics of the model physics and stratospheric chemistry scheme (Richter et al., 2017). For example, while UKESM1-0-LL, CNRM-180 ESM2-1, and IPSL-CM6A-LR show a similar increase in global AOD, their maximum stratospheric temperature increase, and the heating distribution differs substantially (Figure 1, Figure 2a and b). The stratospheric heating per AOD (Figure 2c) is largest for the MPI models, followed by IPSL-CM6A-LR. CESM2-WACCM6 shows a slightly smaller peak of the temperature increase. However, the heating in CESM2-WACCM6 extends toward higher altitudes, which may be due in part to injection at higher altitudes than in the other models.

The temperature increase between 30 and 100 hPa and 20°N-20°S reaches between 5 and 13 degrees K for the six different models (Figure 2c). For the following analyses, we only consider the three models with interactive chemistry (CNRM-ESM2-1, UKESM1-0-LL, and CESM2-WACCM6). These show a smaller range of temperature increase, between 5 and 7 degrees K by the end of the century. In contrast to G6sulfur, solar dimming in G6solar does not lead to a significant temperature change

**Figure 2.** a) Evolution of global mean increase in stratospheric aerosol optical depth for each model in G6sulfur b) Evolution of yearly mean stratospheric temperatures (20°N-20°S, 30-100 hPa) for each model in G6sulfur c) Correlation between values in panels a) and b). Both MPI-ESM1-2-LR and MPI-ESM1-2-HR use the same described aerosol distribution.





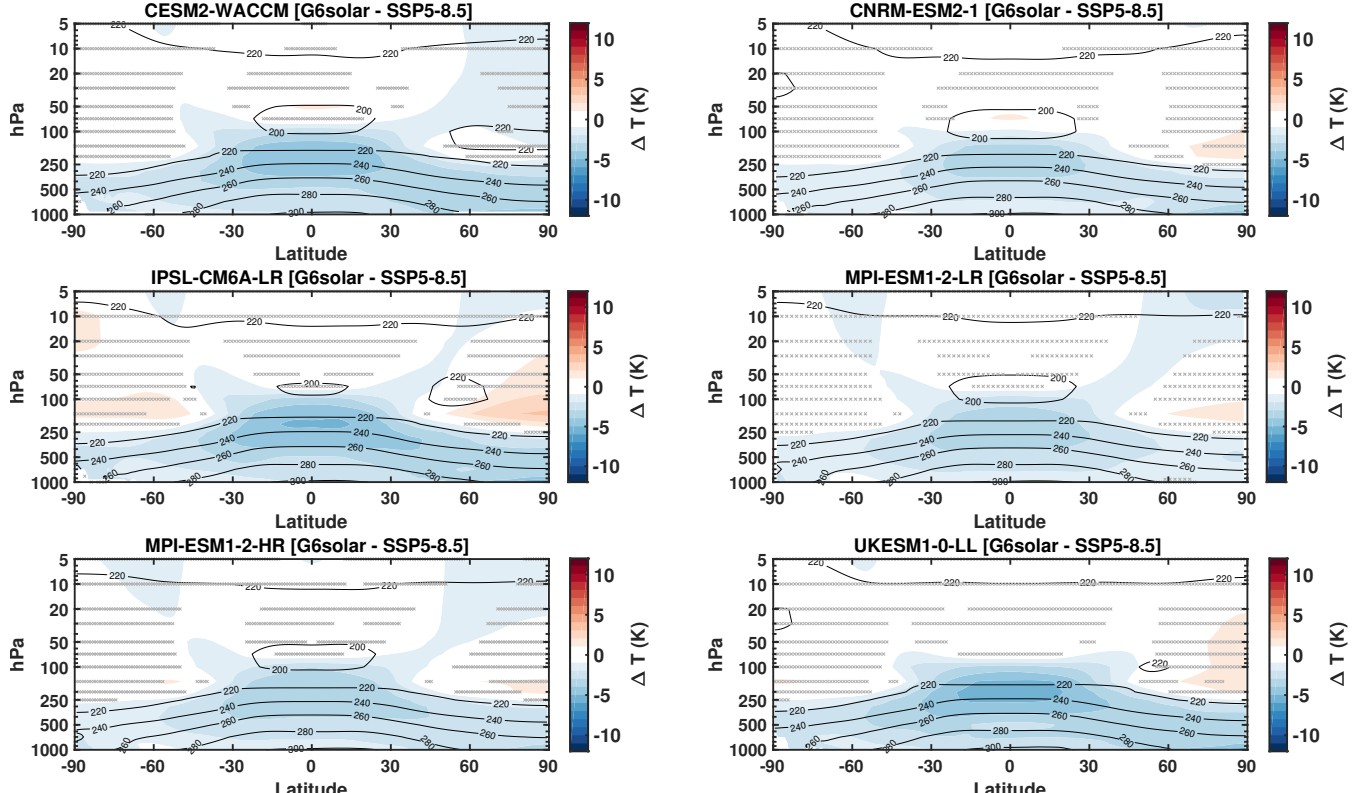

**Figure 3.** Zonal mean temperature changes (2080-99) between G6solar and SSP5-8.5 for all GeoMIP models participating in the G6 experiment. Contour lines show the baseline SSP5-8.5 temperature. Grey horizontal lines indicate changes that are not statistically significant over the time period considered using a double-sided t-test at 95% confidence levels.

in the stratosphere compared to SSP5-8.5 (Figure 3) and stratospheric temperatures stay lower compared to SSP2-4.5. Per
experimental design, the solar dimming in both experiments leads to a similar surface cooling.

The cooling of the troposphere with solar dimming in G6solar results in a slowing of the BDC and weakening of the subtropical jet stream (STJ) and the polar vortex compared to SSP5-8.5 (Figure 4, right column). This experiment therefore successfully reverses the effect of increasing greenhouse gases between SSP2-4.5 and SSP5-8.5 by the end of the century (Figure A1, right column). On the other hand, significant zonal wind changes by the end of the century occur for G6sulfur
(Figure 4, left column) for all three models, including a weakening of the subtropical jet stream and a strengthening of the polar vortex compared to SSP5-8.5, consistent with what has been found in early studies (e.g., Tilmes et al., 2017; Jones et al., 2021).

Changes in the tropical upwelling, illustrated by the vertical wind component ($w^*$) derived from the Transformed Eulerian Mean TEM stream function, depend on the details of the experiment (Figure 5). Only WACCM6 results are analyzed due to
the lack of available information for the other models. For G6sulfur, the weakening of the sub-tropical storm tracks is aligned





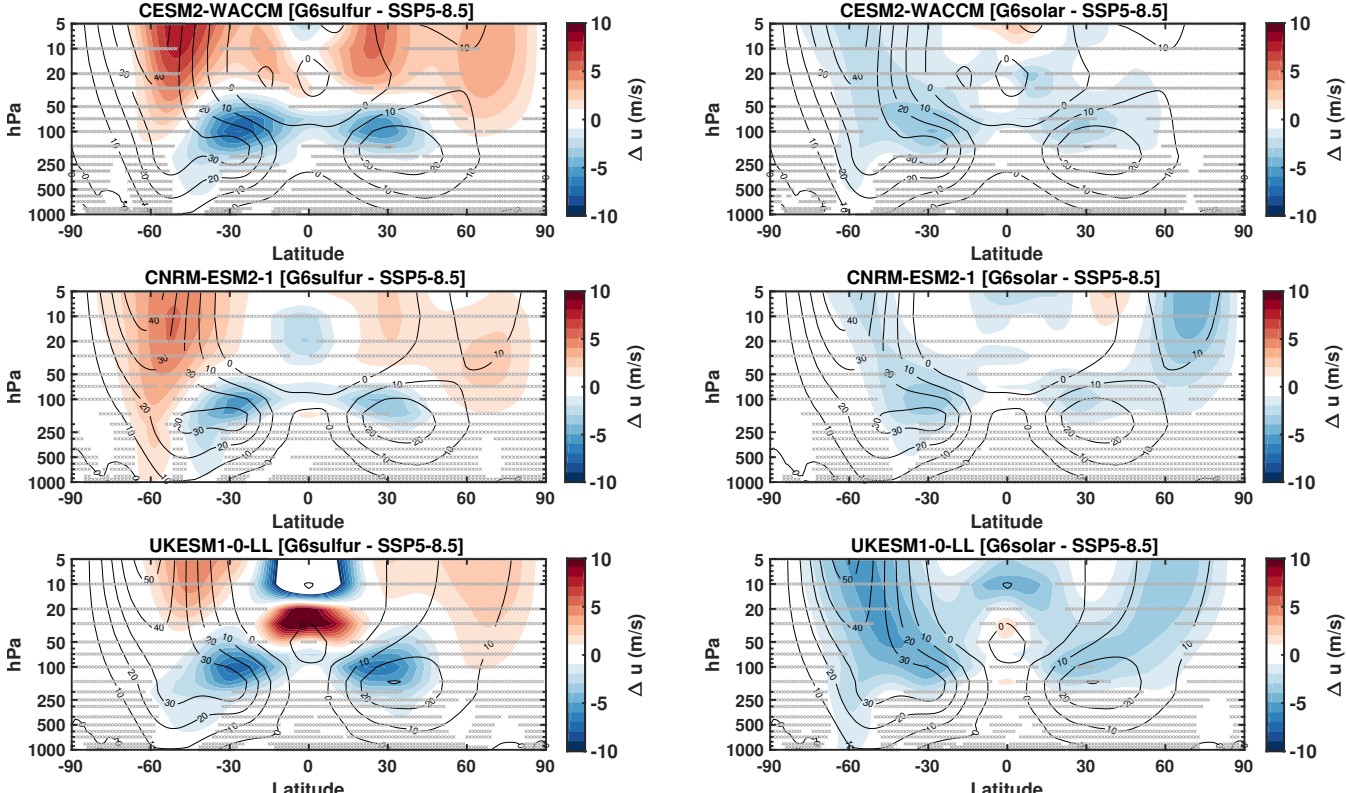

**Figure 4.** Zonal mean zonal winds changes (m/s) (2080-89) between G6sulfur and SSP5-8.5 (left) and G6solar and SSP5-8.5 (left) for the three GeoMIP models with interactive chemistry that participated in the G6 experiment. Contour lines show the baseline SSP5-8.5 winds. Grey horizontal lines indicate changes that are not statistically significant over the period considered using a double sided t-test at 95% confidence levels.

with reduced vertical wind velocity around the tropopause and below the injection altitude. Interestingly, the reduction in $w^*$ overcompensates conditions for the target simulation and results in values similar to present-day conditions. Above the sulfur injection location, $w^*$ is increased compared to SSP5-8.5. In contrast, $w^*$ in G6solar matches the target scenario SSP2-4.5. This indicates that changes in $w^*$ in G6solar are largely driven by tropospheric temperatures.

Besides the commonalities among the three models, some differences exist. For the G6sulfur experiments, the three models differ in their response in zonal wind between 50 and 5 hPa (Figure 4). WACCM6 shows a strengthening of the tropical winds and CNRM-ESM2-1 shows a weakening of the tropical winds. UKESM1-0-LL shows a strong increase of the zonal tropical winds between 20-50 hPa and a decrease above those altitudes by the end of the century which is aligned with the permanent lock-in of the QBO into a westerly phase after 2055 (Jones et al., 2021). These differences are likely connected

to the differences in the heating response that are caused by different injection strategies with injections in lower altitudes showing stronger heating and change in $w^*$ than injections in higher altitudes (Tilmes et al., 2017).

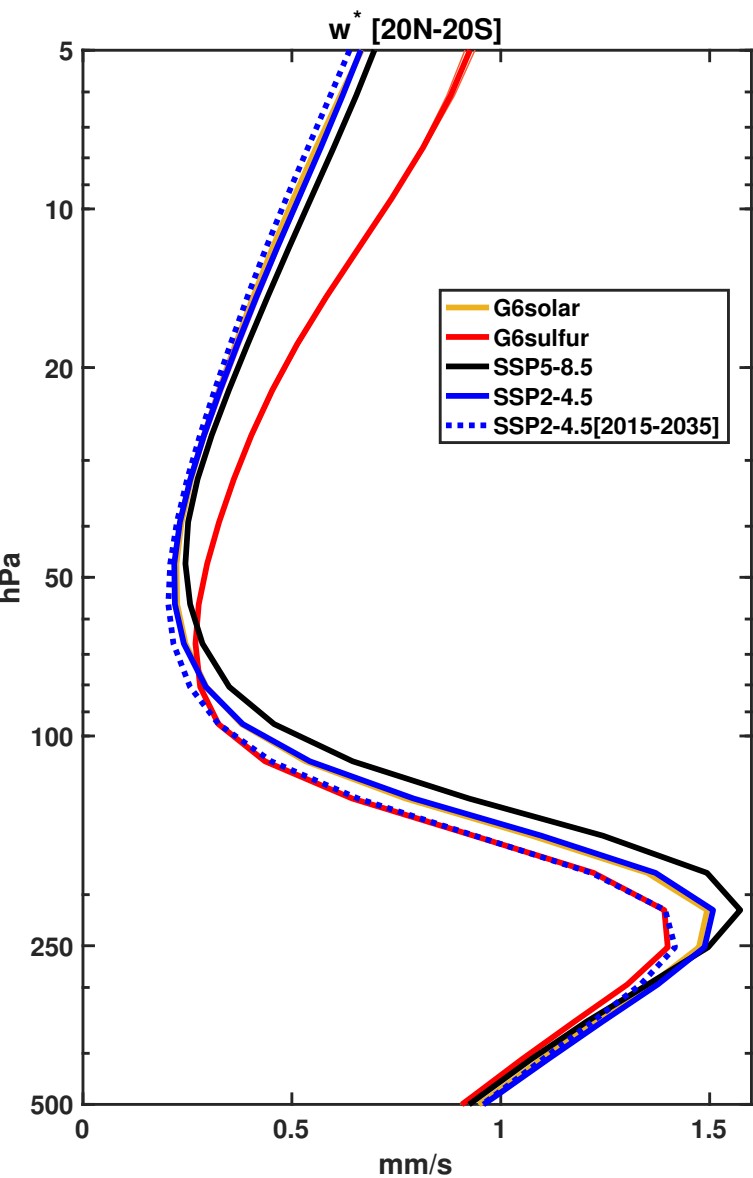

**Figure 5.** Zonally and annually-averaged residual vertical velocity ($w^*$), averaged between $30°$N and $30°$S for CESM2-WACCM6 for 2080-2100 and for different experiments (colored solid lines) and for 2015-2035 for SSP2-4.5 (blue dotted lines).





Furthermore, some non-significant differences in the model results for both G6sulfur and G6solar are obvious for the response already in the first 20 years of the application (Figure A3). CNRM-ESM2-1 shows a weakening of the polar vortex in the Southern Hemisphere for G6sulfur and a strengthening of the polar vortex in both hemispheres in G6solar compared to
the baseline simulation. This, however, is not likely related to the solar or sulfur applications because they had not ramped up before 2040.

### 3.2 Effects of SAI on Surface Area Density

The three ESMs with interactive chemistry applied different strategies to counter the warming between SSP5-8.5 and SSP2-4.5 with stratospheric aerosols (see Section 2). Resulting differences in the changes in SAD as described in this section (Figure 6)
have different impacts on heterogeneous chemistry and, therefore, ozone in the stratosphere.

As described in Section 2, CESM2-WACCM6 injected sulfur at 25 km ($\approx$30 hPa). This resulted in an aerosol distribution that covers a larger altitude range compared to that of other models. Furthermore, the experiment required an initial increase of sulfur emissions of around 2 TgSO$_2$/yr in the first three years of the start of the application, which stayed around 2-3 TgSO$_2$/yr emissions until 2045 (as shown in Visioni et al. (2021b)). This relatively small injection amount results in a sudden increase of
SAD from 2 to 10 $\mu$m$^2$/cm$^3$ within the first year of the application (Figure 6). This happens because the aerosol microphysical scheme first produces smaller particles that grow slowly with increasing injection, resulting in the initial increase in SAD (as also discussed in Tilmes et al. (2021)). After this initial increase, SAD and AOD (Figure 2a) stay constant until about 2050 along with the roughly constant injection amount. After 2050, increased sulfur emissions required to counter the increasing warming resulted in a moderate increase of SAD in the tropics (Figure 6f). The increase in SAD differs among regions and
seasons. In winter and spring, the northern hemisphere mid and high latitudes see a stronger increase in SAD than the SH polar regions and the tropics. A possible reason is the increased meridional transport of air and aerosols towards the northern hemisphere mid and high latitudes with SAI applications, which can be caused by the weakening of the subtropical jet stream that reduces the subtropical wave forcing and therefore decelerates the shallow branch of the BDC.

UKESM1-0-LL injected sulfur in 18-20 km and between 10°N-10°S, as originally specified by the G6sulfur protocol in
Kravitz et al. (2016). The resulting aerosol distribution is nonetheless constrained in a small region above the tropopause, leading to a large peak in SAD (above 70 $\mu$m$^2$/cm$^3$) between 10°N-10°S and between 100-50 hPa by the end of the century, and smaller SAD outside that region compared to the other models (Figure 6c). Sulfur emissions for G6sulfur started in 2030 and ramped up every 10 years with a small increase until 2050 and accelerated increases after that. After 2050 SAD follows a similar trend in the tropics to that in CESM2-WACCM6 (Figure 6f). However, the initial strong increase in SAD found in
CESM2-WACCM6 does not occur in UKESM1-0-LL. This could be a result of the production of larger aerosol particles (and therefore smaller SAD) in UKESM1-0-LL compared to CESM2-WACCM6 for the same injection amount because of stronger vertical transport in CESM2-WACCM and stronger confinement of the enhanced aerosol distribution in UKESM1-0-LL. Long-term increases in SAD are similar to CESM2-WACCM6 in the tropics after 2040, but much lower than the other two models in high latitudes, particularly in the SH.





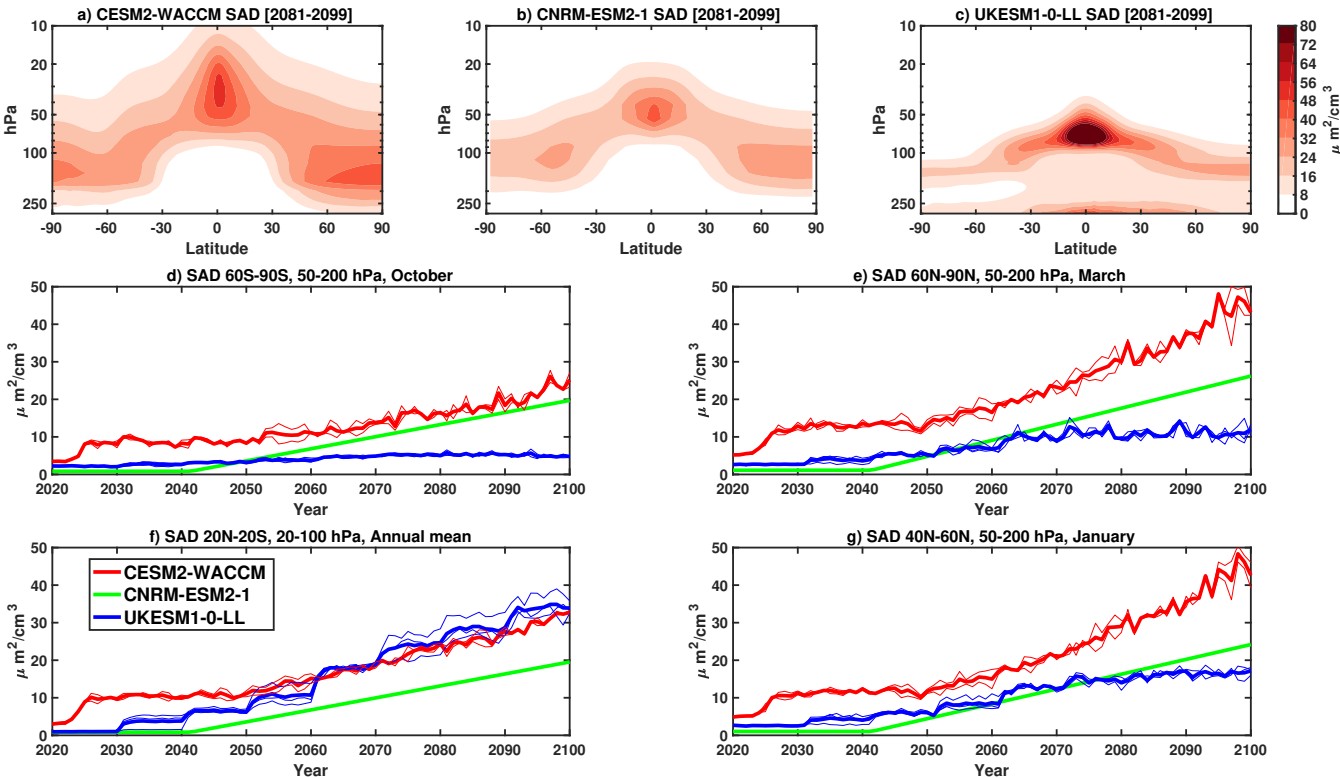

**Figure 6.** a-c) Surface Area Density (SAD) ($\mu$m$^2$/cm$^3$) for the three models in the 2081-2099 period. d-g) Temporal evolution of SAD for the three models over the whole simulation period for different locations and time of the year; d) 60°S-90°S, October e) 20°N-20°S, Annual mean, f) 40°N-60°N, January, g) 60°N-90°N, March.

CNRM-ESM2-1 uses a prescribed stratospheric aerosol distribution that is scaled depending on the requirement to offset the warming between SSP2-4.5 and SSP5-8.5. The aerosol and SAD distribution are generally similar to WACCM6 but smaller and slightly less spread out by the end of the century (Figure 6b). CNRM-ESM2-1 does not apply SAI until after 2040 and applies a linear increase with time after that date mainly because the difference in surface air temperature between SSP8-8.5 and SSP2-4.5 cannot be disentangled from the model internal variability before 2040 (Visioni et al., 2021b). We find a linear increase

with time in SAD because of the scaled, fixed aerosol distribution that does not consider potential changes in the aerosol size distribution with injection amount, or changes in the spatial distribution of the aerosol from transport. The resulting SAD is smaller (by almost half) in the Tropics compared to the other two models. A similar but slightly smaller increase after 2060 is found in the SH high latitudes in October, half the increase in SAD in the NH high latitudes in March, a similar SAD in NH mid-latitudes in January compared to UKESM1-0-LL.





## 3.3 Effects of SAI and solar dimming on ozone concentration

As for the temperature response, first we briefly outline differences in ozone mixing ratios between SSP2-4.5, and SSP5-8.5 (Figure A4). The increasing acceleration of the BDC (Section 3.1) results in ozone reduction around the tropical tropopause due to the increased transport of ozone-poor air masses into the lower tropical stratosphere. In addition, the cooling of the stratosphere results in a slowing of temperature-dependent catalytic ozone loss reactions, which causes an increase of ozone

under colder stratospheric conditions (Haigh and Pyle, 1982), as further outlined in Nowack et al. (2016).

Solar dimming reverses the acceleration of the BDC for G6solar. This results in an increase of ozone in the lower tropical stratosphere and a decrease in the upper troposphere and lower stratosphere (UTLS) at middle and high latitudes (for WACCM6 and CNRM-ESM-1), as also found in Nowack et al. (2016) and Xia et al. (2017). However, solar dimming does not reverse the increase (super-recovery) of ozone in all of the stratosphere. In contrast, it can result in a further ozone increase in the upper

stratosphere compared to the baseline scenario, as most apparent in UKESM1-0-LL (Figure 7, lower right panel). As explained in Nowack et al. (2016), this is based on two main drivers: firstly, the reduced insolation in G6solar results in less ozone photolysis and less abundant atomic oxygen and with that a slowing of catalytic ozone loss reactions, and secondly, a significant reduction in humidity produced by the tropospheric cooling reduces ozone loss via odd hydrogen catalytic cycles. This increase in stratospheric ozone compared to the baseline simulations further impacts ozone in the UTLS and the troposphere through

the exchange of air masses from the stratosphere to the troposphere. The increase in ozone in the UTLS further results in a decrease in oxygen photolysis and therefore an increase in tropospheric ozone.

G6sulfur simulations show a much more substantial increase in ozone right above the tropical tropopause than G6solar for all the models. In addition, results show a reduction of ozone between 50 and 20 hPa, and an increase at about 10 hPa for WACCM6 and UKESM1-0-LL as a result of changes in $w^*$ (Figure 5), as also shown in earlier studies discussed above. In

addition to the dynamical changes, the increase in SAD results in a reduction in the $NO_x$ chemical cycle and an increase in ozone. The increase of ozone around 5-10 hPa is strongest in CESM2-WACCM6, consistent with the largest increase in SAD in that region. This is also consistent with the largest increase of ozone above 50 hPa in mid-and high latitudes by the end of the century in CESM2-WACCM6. In addition, reductions in shortwave radiation as the result of the ozone increase can result in an inverse self-healing of ozone below (Nowack et al., 2016; Pitari et al., 2014), and therefore a reduction in ozone for both

G6sulfur and G6solar, which likely contributes to the reductions in ozone in the Tropics around 20 hPa and in the troposphere for both WACCM6 and UKESM1-0-LL.

Reduction in the strength of the subtropical jet (Figure 4), which are much more pronounced in G6sulfur than in G6solar, are expected to impact the meridional transport of ozone from the tropics to the mid-and high latitudes (as discussed above). The much stronger increase of ozone in the mid-to high latitudes in the UTLS for G6sulfur compared to G6solar (Figure 7) is

aligned with a stronger meridional transport and is most obvious for CNRM-ESM2-1 and UKESM1-0-LL. Ozone depletion in high polar latitudes as a result of increased SAD is most obvious in CESM2-WACCM6 (Figure 7, top panels). As discussed above, smaller SAD in the other models does result in less ozone depletion in these regions. In the case of UKESM1-0-LL,




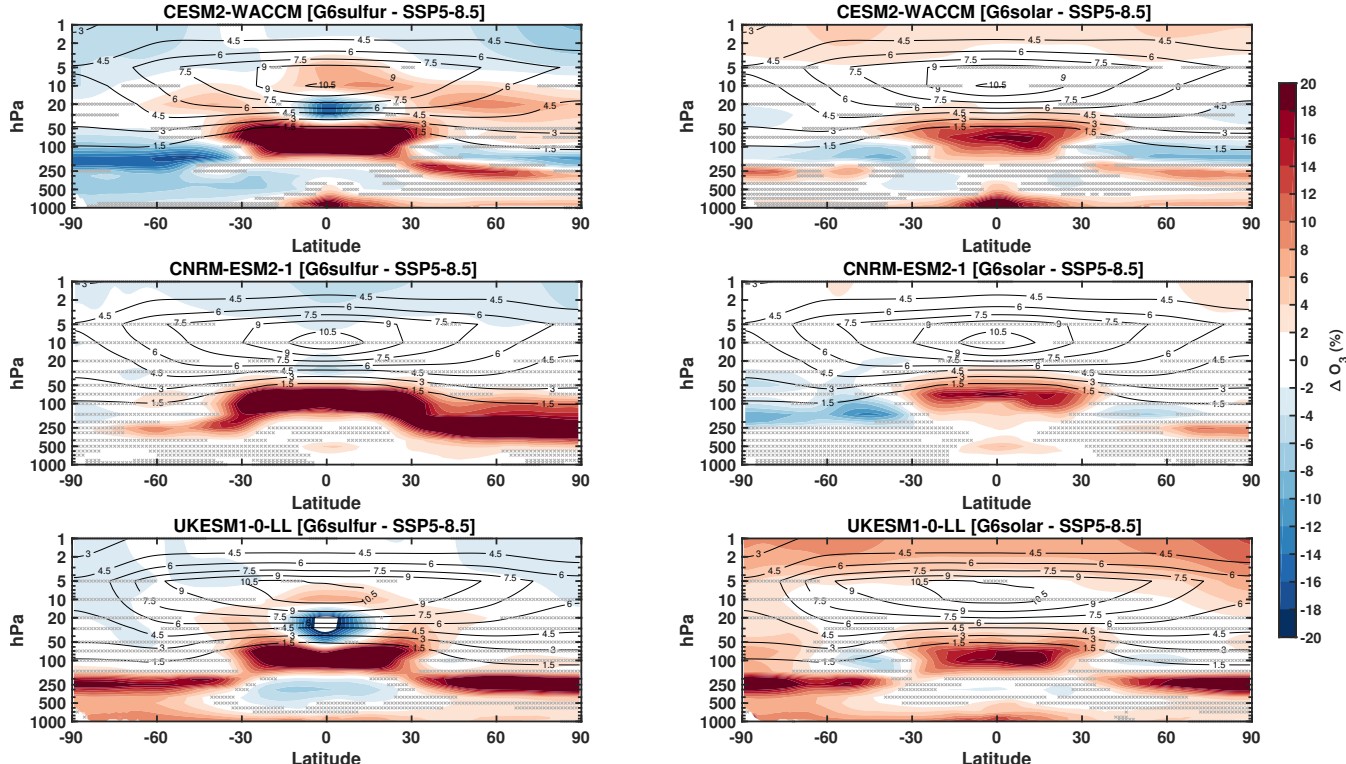

**Figure 7.** Ozone concentration changes (in % of SSP5-8.5) for the G6sulfur (left panels) and G6solar (right panels) cases compared to SSP5-8.5 in the 2080-2099 period. Contour lines show the baseline SSP5-8.5 ozone concentration over the same period. Grey horizontal lines indicate changes that are not statistically significant over the period considered using a double-sided t-test at 95% confidence levels.

heterogeneous activation of halogens on sulfate aerosols does not include any bromine reactions or the important hydrogen chloride plus chlorine nitrate reaction.

Change in ozone in the UTLS further impacts ozone in the troposphere through vertical transport, as also discussed in Xia et al. (2017). WACCM6 shows a reduction in ozone in the lowermost stratosphere (below about 100 hPa) for mid and high latitudes driven by ozone depletion in the polar lower stratosphere. In contrast, the increase of ozone in the stratosphere likely contributes to the increase in ozone in the UTLS for UKESM1-0-LL and CNRM-ESM2-1. In addition, tropospheric ozone is impacted by changes in the photochemistry in both G6sulfur and G6solar, as outlined in detail in Nowack et al. (2016) and Xia

et al. (2017). An increase in tropospheric ozone results from decreased chemical ozone loss due to reduced tropical humidity, resulting from the relative cooling of the surface.

Finally, change in the $O_2$ photolysis rate and UV-B radiation due to changes in ozone and aerosols impacts tropospheric ozone. Reductions in column ozone and the resulting increase in UV-B in high latitudes are partly offset by the reduction in UV-B from the aerosol layer (e.g., Tilmes et al., 2012; Pitari et al., 2014). In this study, UKESM1-0-LL is the only model that

includes an interactive photolysis scheme that takes the effects of aerosols into account, while all the models include changes





due to ozone. The increase in aerosol burden and the resulting reduction of oxygen photolysis likely contribute to the increase in tropospheric ozone in UKESM1-0-LL.

### 3.4 Effects of SAI and solar dimming on total column ozone (TCO)

Total column ozone in SSP5.8.5 and SSP2-4.5 increases in mid-and high latitudes between 2020 and 2100 and reaches above
1960 values for high greenhouse gas forcing scenarios due to the slow reduction in stratospheric halogen loading as the result of the Montreal Protocol (WMO, 2018; Keeble et al., 2021). In addition, enhanced greenhouse gases cool the stratosphere and in turn slow down ozone-destroying reactions, resulting in an increase of ozone. In addition, a warmer troposphere drives the acceleration of the BDC and concomitant changes in the stratospheric lifetime of tracers, including ozone (WMO 2018). On the other hand, in the tropics, high forcing scenarios show an initial increase and later a decrease in TCO, consistent with an
acceleration of the BDC and a decrease in the tropical lower stratospheric ozone (e.g., Meul et al., 2016; Keeble et al., 2017).

The three GeoMIP models follow the general behavior outlined above; however, specific differences exist (Figure 8, black and blue lines). Both WACCM6 and CNRM-ESM2-1 show a stronger recovery in SSP5-8.5 than in SSP2-4.5 for spring in high polar regions, which is in agreement with the multi-model mean derived in Keeble et al. (2021). However, UKESM1-0-LL does not show significant changes in TCO between these two forcing scenarios. On the other hand, WACCM6 does not show a very
strong recovery in the NH high polar regions (Gettelman et al., 2019).

### 3.4.1 Effects of SAI on TCO

In the SH polar region in October, for G6sulfur compared to SSP5-8.5 (Figure 9), WACCM6 shows a significant decline in TCO up to 30 DU for the ensemble mean at the start of the sulfur injection in 2020. After that, TCO declines much slower towards 38 DU by the end of the century. The changes are aligned with changes in SAD (Figure 6d), since chemical changes strongly
control the ozone in this region as well as the slow decline in stratospheric halogen content resulting in reduce chemical ozone loss. CNRM-ESM2-1 simulates decreasing TCO between 2040 and 2100, which is also aligned with the increase in SAD and is smaller than what is simulated in CESM2-WACCM6. However, due to the linear increase in SAD, CMRM-ESM2-1 does not show a strong decrease in ozone during the onset of the SAI application. UKESM1-0-LL shows much smaller reductions in TCO in the SH polar region than the other models due to a smaller increase in SAD. Because of differences in timing and
magnitude of SAD changes, there is a large spread in the TCO response between the three models in this region. The ensemble mean shows an initial decrease in TCO of 10 DU ozone loss and closer to 20 DU by the end of the century (Figure 11). Compared to SSP2-4.5, there is no significant change in TCO besides the initial reduction.

TCO in the NH polar region is strongly controlled by the dynamical variability for different years in addition to chemical changes. WACCM6 does not show any significant changes in TCO, while CNRM-ESM2-1 shows a reduction in TCO in the
first 40 years. Despite no changes in SAD until 2040, this reduction is consistent with the strengthening of the polar vortex in G6sulfur compared to SSP5-8.5. UKESM1-0-LL reproduces an initial reduction in TCO of up to 30 DU by the onset of SAD in 2030. Based on the multi-model mean (Figure 11, right column) differences of G6sulfur compared to SSP5-8.5 only show







**Figure 8.** Ensemble mean of Total Column Ozone evolution between 2020 and 2100 normalized to 2020 values for different experiments (different colors) and four different regions and seasons (different rows) and three models (different columns). The 2-sigma standard deviation of the ensemble mean is only shown for SSP5-8.5 and SSP2-4.5 scenarios for better readability.





**Figure 9.** Differences between G6sulfur and SSP5-8.5 for the ensemble mean of Total Column Ozone between 2020 and 2100 for the three different models (colored lines) and for four different seasons and regions (different panels).

an initial decrease by the onset of SAI and a small (below 20 DU) increase in ozone towards the end of the century. Compared to SSP2-4.5, TCO shows a substantial increase up to 40 DU by the end of the century.

Similar to the NH polar region in March, dynamics and transport strongly impact NH mid-latitudes in winter. A weakening of the STJ results in enhanced meridional transport of air masses towards mid-and high latitudes in G6sulfur compared to SSP5-8.5. All models show a consistent increase in TCO up to 20 DU by the end of the century. WACCM6 shows a stronger initial increase, likely because of the earlier start of SAI. Compared to SSP2-4.5, the multi-model mean reaches above 40 DU. The very robust increase in TCO correlates with the amount of sulfur injections.

The impact of G6sulfur on TCO in the tropics shows a mixed signal (Figure 8, bottom row, and Figure 9). Both CNRM-ESM2-1 and WACCM6 describe an increase in TCO with SAI, while UKESM1-0-LL shows a decrease compared to SSP5-8.5.





Ozone concentrations are increasing around the tropopause for all three models. Both CESM2-WACCM6 and UKESM1-0-LL show a decrease in ozone around 20 hPa, most pronounced in UKESM1-0-LL. This is likely driven by the increase in $w^*$ as discussed above and could have a larger effect than the increase in ozone in the upper stratosphere, therefore resulting in a net

decrease in TCO in UKESM1-0-LL. In addition, CESM2-WACCM6 shows a stronger increase in ozone above 20 hPa aligned with an increase in SAD in that region, as discussed above. The multi-model mean in the tropics shows non-significant changes between G6sulfur and SSP5-8.5 and an increase in TCO around 2 DU compared to SSP2-4.5.

### 3.4.2 Effects of solar dimming on TCO

Change in ozone in G6solar compared to SSP5-8.5 are impacted by changes in transport (including $w^*$) comparable to SSP2-

4.5 conditions, while stratospheric temperatures remain mostly unchanged. In addition, as discussed above, the reduced ozone photolysis due to changes in insolation and reductions in humidity due to the cooling of the troposphere can increase upper atmospheric and tropospheric ozone.

For the SH polar region in October minor changes in ozone are simulated in all models in the first half of the $21^{st}$ century (Figure 10). Only UKESM1-0-LL simulates an increase in TCO of 20 DU compared to SSP5-8.5 in the last 20 years of the

applications. Differences with respect to the other models are likely caused by including the effects of aerosol loading on the photolysis calculation. For the NH polar region in March, CNRM-ESM2-1 shows a decrease of TCO between 2040 and 2080 and an increase after that. UKESM1-0-LL simulates an increase of TCO after 2080, and WACCM6 shows a decrease by the end of the century. Given the large variability in the NH polar region in March, these changes may not be significant. The multi-model mean in high polar latitudes in spring (Figure 11) shows only a minor increase in TCO below 10 DU by the end

of the $21^{st}$ century compared to SSP5-8.5. However, compared to SSP2-4.5, ozone shows a significant increase up to 30 DU for both hemispheres by the end of the $21^{st}$ century.

For NH mid-latitudes in January, solar dimming has no significant effect for any of the models. However, for the Tropics, solar dimming results in a consistent increase between 4 and 8 DU for the three models compared to SSP5.8.5 and over 8 DU compared to SSP2-4.5 in the multi-model mean. This is likely the result of the deceleration of the BDC upwelling.

## 4  Summary

We used the GeoMIP experiments G6sulfur and G6solar to identify the impacts of SAI and solar dimming on stratospheric ozone. The results from the only three ESM with comprehensive stratospheric chemistry were used. The G6sulfur and G6solar baseline experiment employ the high climate forcing scenario SSP5-8.5. SAI and solar dimming are applied to reach surface temperatures of the SSP2-4.5 target experiment. For the analysis of the results, we used limited model output, including zonal-

mean temperature, zonal winds, aerosol surface area density, and ozone. Some additional quantities, including the vertical component of the TEM circulation, $w^*$, were only derived for one model (CESM2-WACCM6).

Both G6solar and G6sulfur applications result in significant changes compared to SSP5-8.5 and SSP2-4.5 with regard to TCO, which differ by region and season. Both model experiments include, per design, reductions of solar radiation reaching





**Figure 10.** Differences between G6solar and SSP5-8.5 for the ensemble mean of Total Column Ozone between 2020 and 2100 for the three different models (colored lines) and for four different seasons and regions (different panels).

**Figure 11.** Left column: Multi-model mean of Total Column Ozone evolution between 2020 and 2100 normalized to 2020 values for different experiments (different colors) and four different regions and seasons (different rows). The 2-sigma standard deviation of the ensemble mean is only shown for SSP5-8.5 and SSP2-4.5 scenarios for better readability. Right column: Differences between G6sulfur and SSP5-8.5 (black lines), G6sulfur and SSP2-4.5 (blue lines), G6solar and SSP5-8.5 (brown lines), and G6solar and SSP2-4.5 (green lines), for the multi-mean of Total Column Ozone between 2020 and 2100 for four different seasons and regions (different rows).





the ground (due to the reflection of short-wave radiation by aerosols or to solar dimming), and a decrease of tropospheric

temperatures to SSP2-4.5 conditions. Significant differences between the two approaches include the increase in absorbing sulfate aerosols in the stratosphere in G6sulfur, which increases lower tropical stratospheric temperatures and stratospheric transport, and the increase in aerosol burden and therefore SAD.

For G6sulfur, the temperature increase in the lower tropical stratosphere ranges between 5 to 13 K by the end of the century for the suite of GeoMIP models that performed this experiment, and between 5 to 7 K for those models that include interac-

tive stratospheric chemistry. The heating of the lower tropical stratosphere by the sulfate aerosol causes a weakening of the subtropical jet, a strengthening of the polar vortex, a reduction in tropical upwelling below the injection location of sulfur, and an increase in the tropical upwelling above the injection locations with respect to SSP5-8.5, as already observed in previous simulations (Pitari et al., 2014; Richter et al., 2017; Visioni et al., 2020; Niemeier et al., 2020). For G6solar, stratospheric temperatures stay close to the conditions in the baseline simulation, SSP5-8.5, while tropospheric temperatures, stratospheric

winds, and tropical upwelling are more similar to SSP2-4.5 conditions. In addition to these changes, increasing SADs in G6sulfur compared to G6solar increases chemical production and loss rates due to increasing heterogeneous reactions. An increase of SAD in the high polar latitudes in winter and spring reduces ozone, while increases in the tropics and mid-latitudes mid-and upper stratosphere increase ozone.

The combination of changes described above impacts TCO in G6sulfur and G6solar in addition to changes from increasing

GHGs in SSP5-8.5 and SSP2-4.5. Ozone reduction in G6sulfur under SAI conditions for all three models by the end of the 21$^{st}$ century has been identified only for October in the southern polar latitudes. An initial significant decrease of polar ozone is only simulated in CESM2-WACCM6, because of a strong initial increase in SAD in the polar region between 2020 and 2030. Since the other models required a later start of sulfur injections and may have potential shortcomings in processing ozone loss based on heterogeneous reactions, they produced initially a smaller increase in SAD in high latitudes. The decrease in TCO counters

the super-recovery of TCO in SSP5-8.5 without SAI towards SSP2-4.5 conditions. In northern high latitudes in March, only UKESM1-0-LL shows a strong decrease of TCO at the onset of SAI. The lack of some ozone loss in CESM2-WACCM6 during the onset of SAI may result from too warm NH polar temperatures in this model. In the multi-model mean, TCO shows a small increase of up to 10 DU by the end of the 21$^{st}$ century for this region and season. All models consistently show an increase in TCO in the NH winter mid-altitudes, as the result of meridional transport of more ozone in the lower tropical tropopause

compared to the baseline scenario. On the other hand, the changes in TCO in the tropics are mixed in sign and magnitude among the models, likely because of differences in changes in tropical upwelling, aerosol distribution, and treatment of ozone photolysis.

The impact of solar dimming compared to SAI on ozone is very different in some regions. For October in the SH polar latitudes, one model shows an increase in TCO with increasing solar dimming, while the other two models don't show a

significant change. The response in the NH high latitudes is spring is also mixed and does not point to significant changes compared to SSP5-8.5 conditions. Similarly, TCO shows no significant changes at high and mid latitudes. However, in contrast to G6sulfur all models show a significant increase in TCO in the tropics with increased solar dimming.





# 5   Conclusions

Recent literature often states that SAI would lead to ozone loss (e.g., Keith and Irvine, 2016). Here we analyze three inde-
pendent ESMs and confirm that ozone loss in the high southern latitudes would still be a concern if SAI were to be applied.
However, considering this specific scenario and the multi-model mean, reductions in TCO are rather small and only reach 20
DU compared to SSP5-8.5 and only initial changes in the first two decades compared to SSP2-4.5. The reason is that 2 out of
3 models show no significant ozone loss. Differences in TCO are caused by how much sulfur injections are required to counter
the surface temperature increase between SSP2-4.5 and SSP5-8.5 and how the models represent relevant both physical and
chemical processes. Simplified descriptions of stratospheric aerosols and microphysical schemes may not reflect the increase
in SAD by the onset of SAI. Improvements in models are needed and may change the results significantly.

Models agree that the increase in sulfur injections results in a robust increase in TCO in NH winter middle and high latitudes
up to 20 DU compared to SSP5-85 and up to 40 DU compared to the target scenario SSP2-4.5. This increase in TCO is linearly
related to the increase in sulfur injections and driven by the warming of the tropical lower stratosphere. It would also compound
with the pronounced increase in TCO compared to present-day levels that may reach up to 30 DU on an annual average between
30-60°N and 50 DU between 60-90°N, and that is expected because of the super-recover due to climate change (Dhomse et al.,
2018; Keeble et al., 2021). This large increase in TCO may have potentially large effects on society and ecosystems and would
have to be investigated in detail (Zarnetske et al., 2021). Changes in the tropics using SAI are small and less conclusive based
on the three models.

On the other hand, it has been stated that a less radiatively absorbing material like calcium carbonate would result in less
impact on global ozone and may prevent ozone deletion and other related changes (Masson-Delmotte et al., 2021). However,
this may not be true if one considers solar dimming as an analog to less absorbing or non-absorbing materials. This study
shows that solar dimming would not effectively revert TCO to that of the target experiment SSP2-4.5. In contrast to G6sulfur,
solar dimming and potentially a less absorbing aerosol would not revert the super-recovery of TCO in SSP5-8.5 to SSP2-4.5
conditions in the long run and therefore result in significantly larger TCO values around 30 DU compared to SSP2-4.5 in mid
and high latitudes.

Solar dimming would further significantly increase TCO in the tropics by 4 DU compared to SSP5-8.5 and by up to 8 DU
compared to SSP2-45. However, this change may counter the reductions in TCO around 5 DU for SSP5-85 and SSP2-4.5 by
the end of the $21^{st}$ century due to climate change. Furthermore, the impact of G6sulfur on TCO is in part driven by changes in
photolysis rates. Only one model includes the effects of aerosols on photolysis. This particular model shows in general more
ozone increase with in G6sulfur than the other models. This indicates that improvements in the photolysis scheme in models
are needed in order to improve the simulation of impacts of SAI and solar dimming approaches on ozone.

Finally, the climate intervention scenarios discussed would require a continued increase of sulfur injections well beyond
the $21^{st}$ century in order to keep surface temperatures to SSP2-4.5 conditions and not result in a phase-out of SAI. In addi-
tion, SSP2-4.5 surface temperature conditions do not describe a feasible target to reach the required surface temperature of
1.5°C above pre-industrial conditions in order to prevent significant impacts and potentially reach tipping points. However,



the main finding of the effects of SAI and solar dimming on stratosphere ozone is likely applicable to different lower-forcing experiments.

*Code and data availability.* All data used in this work is available from the Earth System Grid (https://esgf-node.llnl.gov/search/cmip6/)

*Author contributions.* Simone Tilmes performed the analyses and wrote the manuscript. Daniele Visioni performed analyses and helped with the writing process. All the other authors performed the simulations and offered valuable comments on the manuscript.

*Competing interests.* The authors declare no competing interests.

*Acknowledgements.* We thank Douglas Edward Kinnison and Rolando Garcia for helpful comments and suggestions. Support for Daniele Visioni was provided by the Atkinson Center for a Sustainable Future at Cornell University. This work benefited from the French state
aid managed by the ANR under the "Investissements d'avenir" programme with the reference ANR-11-IDEX-0004-17-EURE-0006. Andy Johnes and James Haywood were supported by the Met Office Hadley Centre Climate Programme funded by the UK Government Department for Business, Energy and Industrial Strategy (BEIS). Ulrike Niemeier has been supported by the Deutsche Forschungsgemeinschaft Research Unit VollImpact (FOR2820). Pierre Nabat, Olivier Boucher, and Roland Séférian acknowledge support from the European Union's Horizon 2020 research and innovation programme under grant agreement No 820829 (CONSTRAIN) and 101003536 (ESM2025 – Earth
System Models for the Future). MPI-ESM were performed on the computer of Deutsches Klima Rechenzentrum (DKRZ). The IPSL-CM6 experiments were performed using the HPC resources of TGCC under the allocations 2019-A0060107732 and 2020-A0080107732 (project gencmip6) provided by GENCI (Grand Equipement National de Calcul Intensif). RS and PN thank the support of the team in charge of the CNRM-CM climate model and Meteo-France/DSI supercomputing center which has provided supercomputing time for CNRM-ESM-2 simulations. The CESM project is supported primarily by the National Science Foundation. This material is based upon work supported by
the National Center for Atmospheric Research, which is a major facility sponsored by the NSF under Cooperative Agreement No. 1852977. Computing and data storage resources, including the Cheyenne supercomputer (doi:10.5065/D6RX99HX), were provided by the Computational and Information Systems Laboratory (CISL) at NCAR.



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

## Appendix A: Supporting Figures

This section includes supporting material in the form of additional Figures A1 to A4 as referred to in the main text.





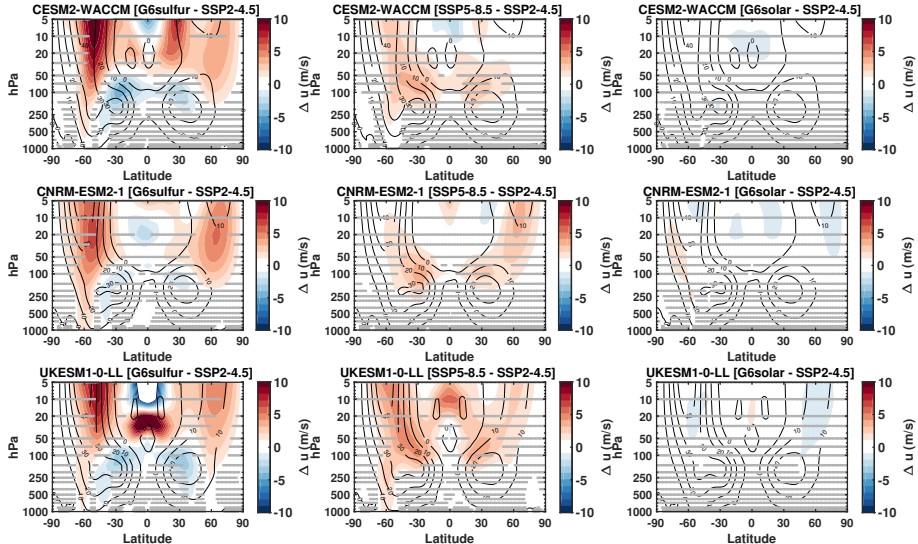

**Figure A1.** Zonal-mean zonal wind changes (m/s) (2080-89) between G6sulfur and SSP2-4.5 (left), SSP5-8.5 and SSP2-4.5 (middle), and G6solar and SSP2-4.5 (right), for the three GeoMIP models with interactive chemistry that participated in the G6 experiment. Contour lines show the baseline SSP2-4.5 winds. Grey horizontal lines indicate changes that are not statistically significant over the temporal period considered using a double-sided t-test at 95% confidence levels.

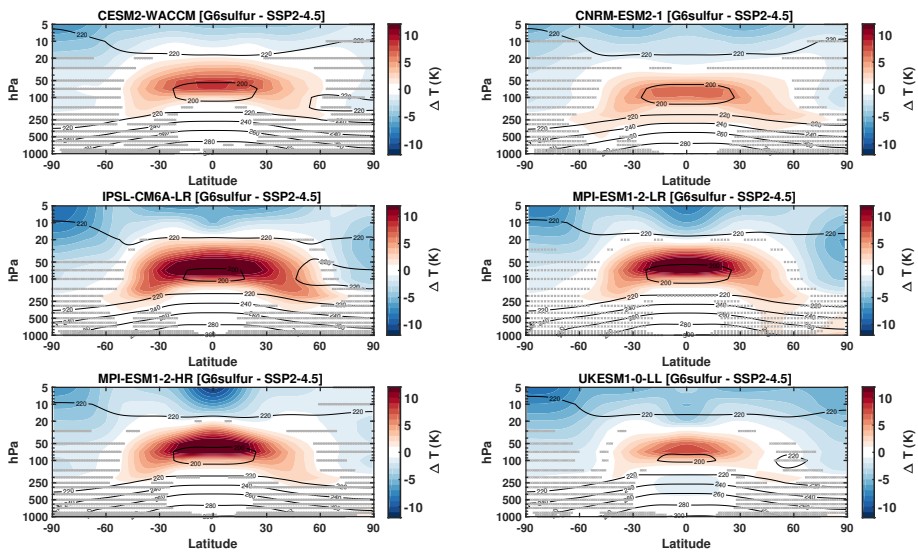

**Figure A2.** Zonal mean temperature changes (2080-99) between G6sulfur and SSP2-4.5 for all GeoMIP models participating in the G6 experiment. Contour lines show the baseline SSP2-4.5 temperature. Grey horizontal lines indicate changes that are not statistically significant over the time period considered using a double-sided t-test at 95% confidence levels.





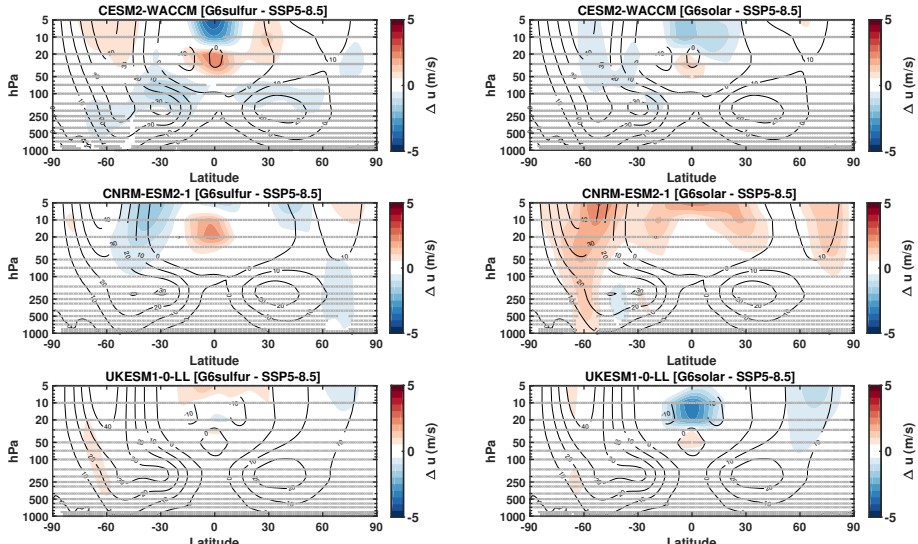

**Figure A3.** Zonal mean U winds changes (m/s) (2030-39) between G6sulfur and SSP5-8.5 (left) and G6solar and SSP5-8.5 (right) for the three GeoMIP models with interactive chemistry that participated in the G6 experiment. Contour lines show the baseline SSP5-8.5 winds. Grey horizontal lines indicate changes that are not statistically significant over the time period considered using a double-sided t-test at 95% confidence levels

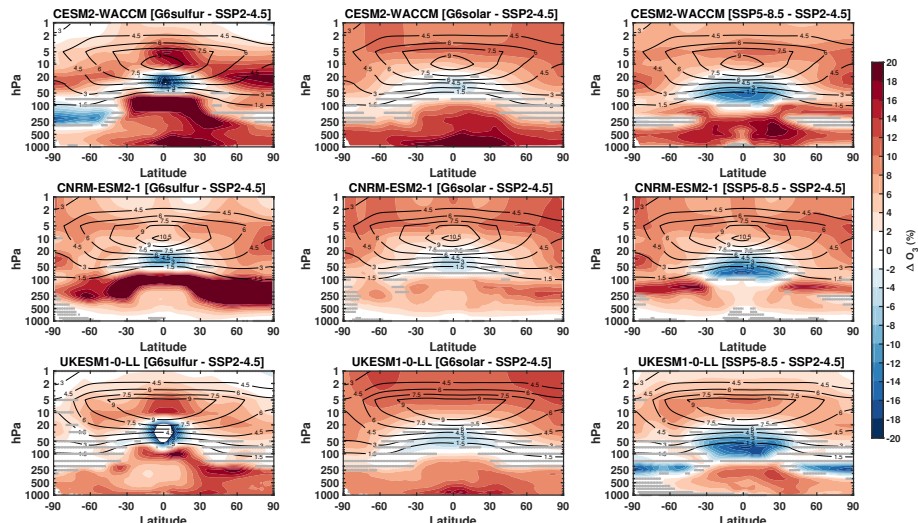

**Figure A4.** Ozone concentration changes (in %) between G6sulfur and SSP2-4.5 (left column), G6solar, and SSP2.-45 (middle column), and SSP5-8.5 and SSP2-4.5 (right panel) in the 2080-2099 period. Contour lines show the SSP2-4.5 ozone concentration over the same period. Grey horizontal lines indicate changes that are not statistically significant over the temporal period considered using a double-sided t-test at 95% confidence levels.