# Peer review of "Stratospheric Ozone Response to Sulfate Aerosol and Solar Dimming Climate Interventions based on the G6 Geoengineering Model Intercomparison Project (GeoMIP) Simulations"

_Atmospheric Chemistry and Physics, 2021_

## Author Response (AR1)

***Response to Reviewer 1 (Ben Kravitz)***

*This is an important study. It is the first to perform a multi-model comparison of the ozone change differences between insolation reduction and stratospheric aerosol injection, which is much needed. The analyses are clear and well described. I'm overall quite pleased with this study. Nevertheless, I am recommending some revisions.*

We thank the Reviewer for the time to comment on the paper and address the comments on a point-by-point basis.

*My biggest concern is that there are some overstatements and loose claims, especially regarding mechanisms. The advantage of a multi-model intercomparison is to learn things you cannot learn from single model studies. If the models are showing vastly different things, you can tie that to individual processes that may or may not be present, albeit doing so can be difficult. If the models are showing similar things (which is often the case here), then you can identify robust mechanisms. My impression of this paper is that sometimes you do this, and sometimes you simply report results and speculate on the mechanisms or reasons for model differences. In the latter case, there is little value added beyond single model studies. I would like to see some more in-depth investigation into why you are getting the results that you are. (See some specific examples below.) I acknowledge that not all models have all of the output you would need to do this. But if the models are showing similar results, you can at least investigate mechanisms in one of the models or cite someone who has already done this.*

We agree with the Reviewer that it is very important to point out the model differences in particular mechanisms and robust features for differences that result in the different answers. While we have tried to do this as much as possible in the paper, we agree that there are some places where we can still add more explanations, as the Reviewer has pointed out. We are addressing those below.

*It's interesting that the main result you found is an increase in TCO, whereas many previous studies found a decrease. I would like to see you talk more about this. What's different between your study and the previous ones?*

It is not new that we find an increase in TCO in the winter Northern Hemisphere latitudes and tropics. We also find a decrease in TCO in the high latitude polar vortex in spring as often stated. In addition, two papers have already discussed this in detail based on CESM-WACCM: 1. Richter at all, 2018, Figure 14, is showing an increase in ozone that is strongest towards the second half of the century (2090-2099) in fall and winter between 40-60N and increases in ozone due to stratospheric aerosol interventions in the Tropics (Figure 1 left). Tilmes et al. (2018) shows this especially for injections closer to the tropopause. Further, other papers (Tilmes et al., 2021) showed the same results based on GLENS (Figure 1, right). There is not really a difference to earlier studies, other than that the focus is on the ozone loss in the polar vortex, which occurs in winter and spring over high polar regions and not so much on other regions and seasons.

To address this comment, we add two references to the sentence below: "Furthermore, the heating of the tropical lower stratosphere results in an increase in total column ozone in mid-to-high latitudes in the Northern Hemisphere winter (**Richter et al., 2018, Tilmes et al., 2018**).

[Figure]

**Figure 14.** Differences in monthly and zonally averaged stratospheric ozone column (in DU) between feedback and RCP8.5 simulations for different months (different panels) as shown in Figure 13, in 2040–2049 (thick lines) and 2090–2099 (dotted lines).

**Figure 4.** Differences of monthly and zonally averaged stratospheric ozone column (in %) between the two sulfur injection experiments and the control simulation averaged between 2042 and 2049, as shown in Figure 3.

Figures by Richter et al., 2018, Tilmes et al., 2018

*You may want to consider moving the appendix to supplemental online material. Appendices count in your page charges, but supplements do not (unless I'm operating under some old assumptions about fee structures).*

Since we only want a few figures that help explain the content of the paper that does not need much additional text. We think that an appendix more appropriate and a supplement is not required.

*Specific comments:*

*Line 8: solar insolation is redundant*

We will change this to: "incoming solar radiation"

*Line 16: Do you mean 2/3 of the models? There are more than 3.*

There are only three models that include interactive chemistry. Two out of the three models show no significant ozone loss in the first 2 decades. We are going to revise the abstract and clarify this point. The new abstract has been updated to make it easier to understand (see the response to Reviewer 2).

*Line 47: denoted*

agreed

*Lines 171-176: I had a lot of trouble with this paragraph. G6sulfur and G6solar contain changes in greenhouse gases. Also, the point of these experiments was not to reverse stratospheric temperature changes in SSP5-8.5, nor would a reversal of those changes be expected.*

We agree with the Reviewer that the sentence was misleading and will rephrase the sentence from:

"G6sulfur and G6solar act in addition to the changes caused by greenhouse gases. The increase Aerosol Optical Depth (AOD) and solar dimming successfully counters the temperature increase in the troposphere and at the global surface between SSP2-4.5 and SSP5-8.5 for all the models (Visioni et al., (2021b). However, stratospheric temperature changes from high greenhouse gas concentrations in the baseline scenario SSP5-8.5 are not reversed. For G6sulfur, the increased stratospheric sulfate burden with increasing sulfur injections causes warming of the lower tropical stratosphere compared to the baseline experiment SSP5-8.5 (Figure 1) and compared to SSP2-4.5 (Figure A2).

to

"G6sulfur and G6solar act in addition to the changes caused by greenhouse gases. The increase **in** Aerosol Optical Depth (AOD) and solar dimming successfully counters the temperature increase in the troposphere and at the global surface between SSP2-4.5 and SSP5-8.5 for all the models (Visioni et al., (2021b). **The cooling of the stratosphere due to the high greenhouse gas concentrations prevails for G6solar.** For G6sulfur, the increased stratospheric sulfate burden with increasing sulfur injections cause warming of the lower tropical stratosphere compared to the baseline experiment SSP5-8.5 (Figure 1) and compared to SSP2-4.5 (Figure A2)."

*Line 178: I was expecting you to talk about the specific differences between the models.*

We agree that it is a good place to mention the reason for differences in the models. Visioni et al. (2021) has already discussed differences in heating between the same models and concluded that more work needs to be done to identify the differences in more detail. In response to this comment, in the revised version, we plan to go a little beyond what Visioni et al. (2021) has discussed and address aspects that may contribute to these differences. First, differences in the radiative scheme play a role, both in the short and the long-wave (Niemeier et al., 2020). We address this by adding information of the radiative schemes used for different models in Section 3 and refer to them in the text later. Second, we point out differences in aerosol distributions, in some models they are prescribed, but different models produce different aerosol distributions and characteristics, that influence the heating. And third, we point out differences in chemistry that can also impact the heating.

To address this comment, we add the following paragraph with information on the radiative schemes, in Section 3:

In addition to differences in chemistry, different radiative schemes contribute to differences in aerosol heating in G6sulfur. The UK radiations scheme is based on the SOCRATES (https://code.metoffice.gov.uk/trac/socrates, last access: 4 April 2019) radiative transfer scheme (Edwards and Slingo, 1996; Manners et al., 2015) with a new configuration for GA7 (Walters et al., 2019). The MPI models use the radiation scheme by Pincus and Stevens (2013) and Mauritsen et al. (2019). This radiation scheme is a modification of the Rapid Radiation Transfer Model (RRTM). Both CNRM-ESM2-1 and IPSL-CM6A-LR use an updated version of the Fouquart-Morcrette scheme (Fouquart and Bonnel, 1980, Morcrette et al., 2008) with 6 bands for the short-wave and 16 bands of the RRTM scheme (Mlawer et al. 1997) for the long-wave radiation. Finally, CESM2-WACCM uses RRTM for both long-wave and short-wave radiation.

We further change the following text:

"Different models substantially differ in the amount of stratospheric heating in G6sulfur by the end of the experiment (2080-2099). This difference results from variations in the amount of aerosol mass and size, aerosol distribution, and the specifics of the model physics and stratospheric chemistry scheme Richter et

al., 2017). For example, while UKESM1-0-LL, CNRM-ESM2-1, and IPSL-CM6A-LR show a similar increase in global AOD, their maximum stratospheric temperature increase, and the heating distribution differs substantially (Figure 1, Figure 2a and b). The stratospheric heating per AOD (Figure 2c) is largest for the MPI models, followed by IPSL-CM6A-LR. CESM2-WACCM6 shows a slightly smaller peak of the temperature increase. However, the heating in CESM2-WACCM6 extends toward higher altitudes, which may be due in part to injection at higher altitudes than in the other models.

The temperature increase between 30 and 100 hPa and $20°N-20°S$ reaches between 5 and 13 degrees K for the six different models (Figure 2c). For the following analyses, we only consider the three models with interactive chemistry (CNRM-ESM2-1, UKESM1-0-LL, and CESM2-WACCM6). These show a smaller range of temperature increase, between 5 and 7 degrees K by the end of the century. "

To:

"Models differ substantially in the amount of stratospheric heating in G6sulfur by the end of the experiment (2080-2099) (Figure 2). As pointed out in Visioni et al. (2021), variations in the heating response to sulfates in the models can be caused by different quantities, including aerosol mass and aerosol size distribution, differences in the heating rates as the result of the different radiative schemes, stratospheric chemical composition, and water vapor. Differences in the radiation scheme play an important role (Neely et al., 2016) for both long and short-wave radiation (Niemeier et al., 2020). While the schemes substantially differ for the MPI-ESM1-2 models and UKESM1-0-LL, the CNRM-ESM2-1, IPSL-CM6A-LR, and CESM2-WACCM all use the same radiative schemes for the long-wave radiation (see Section 3). However, even using similar radiative scheme in the long wave, these three models show differences in the heating response (Figure 2c), which is in part a result of differences in the amount and distribution of aerosol mass. For example, CESM2-WACCM6 heating extends toward higher altitudes, because injections were performed at higher altitudes than in the other models. Other differences include using a prescribed aerosol distribution with fixed aerosol sizes that do not increase with increasing injection amount (CNRM-ESM2-1), vs. interactive aerosols schemes.

Another important difference between all six models is the use of prescribed vs. interactive chemistry. Richter et al. (2017) have shown that stratospheric aerosol injection experiments produce more tropical stratospheric heating if the simulation uses prescribed chemistry rather than interactive chemistry. The temperature increase between 30 and 100 hPa and $20°N-20°S$ reaches between 5 and 13 K for all the six different models (Figure 2c). However, if we only consider the three models with interactive chemistry (as used in the following analysis), CNRM-ESM2-1 (with a prescribed aerosol distribution), and UKESM1-0-LL and CESM2-WACCM6 (with interactive aerosols), these models show a smaller range of temperature increase, between 5 and 7 K by the end of the century, consistent with what has been shown in Richter et al. (2017). More specific model experiments will be needed to quantify the contributions of the different factors that lead to differences in the radiative heating. In contrast to G6sulfur, solar dimming in G6solar does not lead to a significant temperature change in the stratosphere compared to SSP5-8.5 (Figure 3) and stratospheric temperatures stay lower compared to SSP2-4.5. As per the experimental design, the solar dimming in both experiments leads to a similar surface cooling."

*Line 189: I'd like to see more discussion here. The range of uncertainty for models that actually inject SO2 is much smaller than the total range. Can you explain/show why?*

The Reviewer must have assumed that the three models with interactive ozone and with the lower temperature range all inject sulfur. However, that is not the case. Two out of the three models with interactive ozone inject sulfur, while one model prescribes AOD (or SAD). We realize that this needs

clarification and will update the text, accordingly, as addressed outlined in the comment above, second paragraph of the revised text.

*Figure 2c: I know this panel is called correlation, but did you compute a correlation or best fit line?*

To address this, we update the figure and add the correlation coefficient (r2) and the slope (s) and change the figure caption accordingly:

[Figure]

Figure 2: a) Evolution of global mean increase in stratospheric aerosol optical depth for each model in G6sulfur b) Evolution of yearly mean stratospheric temperatures (20°N-20°S, 30-100 hPa) for each model in G6sulfur c) Correlation between values in panels a) and b) with the slope of the fitted linear function and the correlation coefficient (R2) shown in the legend (top left). Both, MPI-ESM1-2-LR and MPI-ESM1-2-HR use the same described aerosol distribution.

*Figure 3: Why does panel (d) have a different way of denoting statistical significance?*

Statistical significance is denoted with "x" in all models. Some models have a coarser horizontal resolution than others, which allows more space between the symbols.

*Line 204: Why? I don't think you've convincingly showed this.*

We agree that this sentence was misleading and unnecessary and will remove it.

*Lines 215-216: I'm puzzled by this sentence. If it's not related to geoengineering then that feature would show up in the baseline simulation. Or it's due to natural variability. Either way, you can't make a claim like this without backing it up.*

Yes, this difference is due to natural variability. As the sentences says: "the sulfur injections have not ramped up before 2040", which is shown in Figure 6e with surface area density not changing before 2040. In the revised version, we will clarify the text accordingly.

"Furthermore, some non-significant differences in the model results for both G6sulfur and G6solar **compared to the baseline** are obvious in the first 20 years of the application (Figure A3). CNRM-ESM2-1 shows a weakening of the polar vortex in the Southern Hemisphere for G6sulfur and a strengthening of the polar vortex in both hemispheres in G6solar compared to the baseline simulation. This, however, is not related to the solar or sulfur applications, because **injections in this model** had not ramped up before 2040 **(see below)** and **therefore is a result of internal variability**."

*Lines 231-234: Possible reason? Did you investigate this or do you have a citation to back this up?*

This is shown in Figure 6. We will refer to this figure in the revised version of the paper.

*Line 235: 2015*

Agreed

*Lines 240ff: Could be? It seems like you could (should) figure this out.*

We can't figure it out more than what has been said, because models did not provide size distribution to proof this.

*Lines 301-302: Can you verify this?*

This cannot be verified based on the available output and information, so we have to be speculative here. Certainly though, photolysis is impacted by aerosols (Pitari et al., 2014) as mentioned in the text.

*Line 304: SSP5-8.5*

agreed

*Lines 311-315: It would be nice if you could say more about this. Why are you getting differences?*
We agree with the Reviewer that we can identify the reasons for this, and update the text as indicated below:

"The three GeoMIP models follow the general behavior outlined above; however, specific differences exist (Figure 8, black and blue lines). Both WACCM6 and CNRM-ESM2-1 show a stronger recovery in SSP5-8.5 than in SSP2-4.5 for spring in high polar **latitudes, which agrees** with the multi-model mean derived in Keeble et al. (2021). **While UKESM1-0-LL does not show significant changes in TCO between these two forcing scenarios, this version of the model does not explicitly treat most of the long-lived ODS important for the ozone recovery (see Section 2), which could contribute to this behavior.** On the other hand, WACCM6 does not show a very strong recovery in the NH high polar regions, **because of a warm bias in the model in this region which did not properly reproduce the reduction of ozone between 1980-2000** (Gettelman et al., 2019).

**New References:**

- Niemeier, U., Richter, J. H., and Tilmes, S.: Differing responses of the quasi-biennial oscillation to artificial $SO_2$ injections in two global models, Atmos. Chem. Phys., 20, 8975–8987, https://doi.org/10.5194/acp-20-8975-2020, 2020.
- Neely III, R.R., Conley, A.J., Vitt, F. and Lamarque, J.F., 2016. A consistent prescription of stratospheric aerosol for both radiation and chemistry in the Community Earth System Model (CESM1). *Geoscientific Model Development*, *9*(7), pp.2459-2470.
- Fouquart, Y. and Bonnel, B.: Computations of solar heating of the earth's atmosphere- A new parameterization, Beiträge zur Physik der Atmosphäre, 53, 35–62, 1980.
- Morcrette, J.-J., Barker, H. W., Cole, J. N. S., Iacono, M. J., and Pincus, R.: Impact of a New Radiation Package, McRad, in the ECMWF Integrated Forecasting System, Monthly Weather Review, 136, 4773–4798, https://doi.org/10.1175/2008MWR2363.1, 2008.
- Walters, D., et al.: The Met Office Unified Model Global Atmosphere 7.0/7.1 and JULES Global Land 7.0 configurations, Geosci. Model Dev., 12, 1909-1963 https://doi.org/10.5194/gmd-12-1909-2019, 2019.
- Edwards, J. M. and Slingo, A.: Studies with a flexible new radiation code. I: Choosing a configuration for a large-scale model, Q. J. Roy. Meteorol. Soc., 122, 689–719, https://doi.org/10.1002/qj.49712253107, 1996.  a, b
- Manners, J., Edwards, J. M., Hill, P., and Thelen, J.-C.: SOCRATES (Suite Of Community RAdiative Transfer codes based on Edwards and Slingo) Technical Guide, Met Office, UK, available at: https://code.metoffice.gov.uk/trac/socrates (last access: 25 October 2017), 2015.
- Pincus, R. and Stevens, B., 2013. Paths to accuracy for radiation parameterizations in atmospheric models. *Journal of Advances in Modeling Earth Systems*, *5*(2), pp.225-233.
- Mlawer, E.J., Taubman, S.J., Brown, P.D., Iacono, M.J. and Clough, S.A., 1997. Radiative transfer for inhomogeneous atmospheres: RRTM, a validated correlated-k model for the longwave. *Journal of Geophysical Research: Atmospheres*, *102*(D14), pp.16663-16682.
- Mauritsen, T., Bader, J., Becker, T., Behrens, J., Bittner, M., Brokopf, R., Brovkin, V., Claussen, M., Crueger, T., Esch, M. and Fast, I., 2019. Developments in the MPI-M Earth System Model version 1.2 (MPI-ESM1. 2) and its response to increasing CO2. *Journal of Advances in Modeling Earth Systems*, *11*(4), pp.998-1038.

Corrected reference:
Tilmes, S., Richter, J. H., Mills, M. J., Kravitz, B., MacMartin, D. G., Garcia, R. R., et al. (2018). Effects of different stratospheric $SO_2$ injection altitudes on stratospheric chemistry and dynamics. *Journal of Geophysical Research: Atmospheres*, 123, 4654– 4673. https://doi.org/10.1002/2017JD028146

***Response to Reviewer 2:***

*The paper by Tilmes et al. considers the stratospheric ozone response to sulphate and solar geoengineering across a set of models participating in GeoMIP. As such, the paper investigates an important potential environmental side effect of solar radiation management geoengineering. The study is well-written and quite comprehensive already in terms of the various impacts and points discussed. I would thus recommend publication subject to a few minor comments listed below.*

We thank the Reviewer for helpful comments and suggestions and address them on a point-by-point basis below.

***Minor comments:***

- *l. 12 on p.1 to l. 26 on p.2: in the abstract are quite complex and contain a lot of highly specific details. Maybe this could be summarised more briefly/in a simpler fashion, especially for the general reader. I think the complexity arises from discussing different regions, simulations, baselines, solar, sulphate and GHG effects, plus multiple variables. I recommend just summarising the overarching messages.*

  We agree with the Reviewer that the abstract was complex and will change it to be less complex. Here, we also consider other comments by the Reviewers.

  "This study assesses the impacts of sulfate aerosol intervention (SAI) and solar dimming on stratospheric ozone based on the G6 Geoengineering Model Intercomparison Project (GeoMIP) experiments, called G6sulfur and G6solar. For G6sulfur, an enhanced stratospheric sulfate aerosol burden reflects some of the incoming solar radiation back into space to cool the surface climate, while for G6solar, the reduction of the global solar constant in the model achieves the same goal. Both experiments use the high emissions scenario SSP5-8.5 as the baseline experiment and define surface temperature from the medium emission scenario SSP2-4.5 as the target. Six Earth System Models (ESMs) performed these experiments, and three out of the six models include interactive stratospheric chemistry. The increase in absorbing sulfate aerosols in the stratosphere results in a heating of the lower tropical stratospheric temperatures by between 5 to 13 K for the six different ESMs, leading to changes in stratospheric transport, water vapor, and other related changes. The increase of the aerosol burden also increases aerosol surface area density, which is important for heterogeneous chemical reactions. The resulting changes in spring-time Antarctic ozone between the G6sulfur and SSP5-8.5, based on the three models with interactive chemistry, include an initial reduction of total column ozone (TCO) of 10 DU (ranging between 0-30DU for the three models) and up to 20 DU (between 10-40 DU) by the end of the century. The relatively small reduction in TCO for the multi-model mean in the first two decades results from variations in the required sulfur injections in the models and differences in the complexity of the chemistry schemes. In contrast, in the Northern Hemisphere (NH) high latitudes, no significant changes can be identified due to the large natural variability in the models, with little change in TCO by the end of the century. However, all three models consistently simulate an increase in TCO in the NH mid-latitudes up to 20 DU compared to SSP5-8.5, in addition to the 20 DU increase resulting from increasing greenhouse gases between SSP2-4.5 and SSP5-8.5. In contrast to G6sulfur, G6solar does not significantly change stratospheric temperatures compared to the baseline simulation. Solar dimming results in little change in TCO compared to SSP5-8.5. Only in the tropics, G6solar results in an increase of TCO of up to 8 DU compared to SSP2-4.5, which may counter the projected reduction in SSP5-8.5.

This work identifies differences in the response of SAI and solar dimming on ozone for three ESMs with interactive chemistry, which are partly due to differences and shortcomings in the complexity of aerosol microphysics, chemistry, and the description of ozone photolysis. It also identifies that solar dimming, if viewed as an analog to SAI using a predominantly scattering aerosol, would succeed in reducing tropospheric and surface temperatures, but any stratospheric changes due to the high forcing greenhouse gas scenario, including the potential harmful increase in TCO beyond historical values, would prevail."

*p. 2 l. 43: maybe the effect of changing TTL temperatures on stratospheric water vapour / composition is also worth mentioning?*

We agree and are going to mention it: "In addition, the heating of the lower tropical stratosphere from sulfate aerosols causes changes in stratospheric transport and circulation, **and an increase in stratospheric water vapor** …"

- *p. 6 l. 168: the effect on the BDC requires a reference*

We add the following reference:  Shepherd, T.G. and McLandress, C., 2011. A robust mechanism for strengthening of the Brewer–Dobson circulation in response to climate change: Critical-layer control of subtropical wave breaking. *Journal of the Atmospheric Sciences*, *68*(4), pp.784-797.

- *Figure 1: is the temperature (T) response (stratospheric warming) due to the aerosols mainly determined by how much warming has to be offset in each model (e.g. how much aerosol has to be injected), or is it a real mixture of factors, or other aspects are more relevant (e.g. radiative heating intensity for a given aerosol increase)? I see you comment on this in the main text, i.e. there are substantial differences. Interesting to see that the interactive chemistry models appear to show smaller T-responses. Coincidence or possibly effects of ozone loss (cooling) and increasing in stratospheric water vapour (cooling, too)? Could you comment on this?*

In fact, the models with interactive chemistry show smaller temperature changes with AOD changes. While there are also differences in the radiative scheme, Richter et al., 2017 have shown that a model with interactive chemistry shows smaller heating, due to changes in ozone. We are addressing this comment and a similar one by the other Reviewer and changing the text:

"Another important difference between all six models is the use of prescribed vs. interactive chemistry.  Richter et al. (2017) have shown that stratospheric aerosol injection experiments produce more tropical stratospheric heating if the simulation uses prescribed chemistry rather than interactive chemistry. The temperature increase between 30 and 100 hPa and 20°N-20°S reaches between 5 and 13 K for all the six different models (Figure 2c). However, if we only consider the three models with interactive chemistry (as used in the following analysis), CNRM-ESM2-1 (with a prescribed aerosol distribution), and UKESM1-0-LL and CESM2-WACCM6 (with interactive aerosols), these models show a smaller range of temperature increase, between 5 and 7 K by the end of the century, consistent with what has been shown in Richter et al. (2017). More specific model experiments will be needed to quantify the contributions of the different factors that lead to differences in the radiative heating.  In contrast to G6sulfur, solar dimming in G6solar does not lead to a significant temperature change in the stratosphere compared to SSP5-8.5 (Figure 3) and stratospheric temperatures stay lower compared to SSP2-4.5. As per the experimental design, the solar dimming in both experiments leads to a similar surface cooling."

- *l. 204: "largely driven by tropospheric temperatures". Maybe dangerous to express this way as it ignores the actual mechanism? (lifting of the tropopause under tropospheric warming and changes in the jet stream affecting wave propagation) I know there are uncertainties in the mechanism but best to rephrase. We certainly wouldn't want new students for example to misunderstand this.*

  We agree with the Reviewer and drop this sentence.

- *l.212-216: could this simply be internal variability? 20 years are fairly short for polar vortex coupled atmosphere-ocean simulations. Statistical significance can give strange results in such cases, especially if p-values are not adjusted for multiple hypothesis testing.*

  We agree and adjust the text:

  "Furthermore, some non-significant differences in the model results for both G6sulfur and G6solar **compared to the baseline** are obvious in the first 20 years of the application (Figure A3). CNRM-ESM2-1 shows a weakening of the polar vortex in the Southern Hemisphere for G6sulfur and a strengthening of the polar vortex in both hemispheres in G6solar compared to the baseline simulation. This, however, is not related to the solar or sulfur applications, because **injections in this model** had not ramped up before 2040 **(see below)** and **therefore is a result of internal variability**."

- *Caption Figure 6: maybe clarify: "for the three INTERACTIVE stratospheric chemistry models" and that this is "Aerosol surface area density".*

  We agree with the Reviewer will add the suggestions to Figure 6.

- *l. 268: do you mean STRATOSPHERIC humidity changes (across all models considered)? Better to clarify.*

  Yes, we add "**stratospheric** humidity" to the revised version of the paper.

- *Figure 8: could you increase the resolution slightly? It looks quite coarse.*

  The Figure seems to look OK – we will ensure that it meets the standards of ACP copy-editors.

- *Figure 9: maybe worth highlighting how significant those changes are relative to historical observed variability (e.g. by a grey shading for one observed sigma around zero)? For example, in the top right subfigure models show vastly different variability characteristics, which leaves the reader to wonder which model is too variable or shows too little variability.*

  It is difficult to compare to observed variability here, because TCO has experienced changing trends over the last 50 years, which is different for different regions. Instead, to address the comment, we are noting in the text that the variability of TCO, especially for the Northern Hemisphere high latitudes in March, is very different for the different models, and a significant trend is not detectable. We add in the text after "TCO in the NH polar region is strongly controlled by the dynamical variability for different years in addition to chemical changes." "Both CNRM-ESM2-1 and UKESM1-0-LL show much larger interannual variably in TCO compared to WACCM, which is more in line with what has been observed (Keeble et al., 2021).

- *l. 404 don't -> do not*

  agreed

- *l. 413: can you say though which model(s) probably performed best / are best suited for the task? The CNRM model seemed strangely unresponsive? Is this a feature also found in Keeble et al. (2021), ACP? There, results were extremely dependent on the model and showed quite clearly that certain chemistry responses could be discounted.*

  It is not possible to discount one or the other model here since all of them have significant shortcomings. We add a couple of sentences to the text to address this comment:

  *"Differences in TCO are caused by how much sulfur injections are required to counter the surface temperature increase between SSP2-4.5 and SSP5-8.5 and how the models represent both relevant physical and chemical processes. Simplified descriptions of stratospheric aerosols and microphysical schemes may not reflect the increase in SAD by the onset of SAI. **In summary, all three models with interactive chemistry show shortcomings in representing different processes properly. For instance, CNRM-ESM2-1 uses a prescribed aerosol distribution that does not reflect changes in the aerosols size with emission amount, UKESM1-0-LL version used includes limited representation of heterogeneous halogen reactions on sulfate aerosols as well as simplified ODS to reflect changes in future ozone, CESM2-WACCM does not reproduce Arctic ozone loss very well, and finally, both CNRM-ESM2-1 and CESM2-WACCM do not include changes in aerosol loading in their photolysis scheme.** Improvements in models are needed and may change the results significantly."*

- *l. 425 -430: this is a central result it seems, and probably worth highlighting even more in the abstract instead of the detailed discussion of certain results? Your final sentence in the abstract already covers this point, but it be clearer/more explicit about the implications and why this questions state-of-the-art thinking in our discipline.*

  As Reviewer 1 pointed out, the intention of SAI is to cool the surface but not counter other changes as the result of climate changes, including the increase of stratospheric ozone. We agree, however, that those effects need to be stated and change the last sentence of the abstract slightly: "It also identifies that solar dimming, if viewed as an analog to SAI using a predominantly scattering aerosol, **would succeed in reducing tropospheric and surface temperatures, but any stratospheric changes due to the high forcing greenhouse gas scenario, including the potential harmful increase in TCO beyond historical values, would prevail**."